# Tailoring polymer electrolyte ionic conductivity for production of low-temperature operating quasi-all-solid-state lithium metal batteries

Zhuo Li[1,4], Rui Yu[1,4], Suting Weng [2], Qinghua Zhang[2], Xuefeng Wang [2,3] ✉ & Xin Guo [1] ✉

The stable operation of lithium-based batteries at low temperatures is critical for applications in cold climates. However, low-temperature operations are plagued by insufficient dynamics in the bulk of the electrolyte and at electrode| electrolyte interfaces. Here, we report a quasi-solid-state polymer electrolyte with an ionic conductivity of $2.2 \times 10^{-4}$ S cm$^{-1}$ at $-20\,°C$. The electrolyte is prepared via in situ polymerization using a 1,3,5-trioxane-based precursor. The polymer-based electrolyte enables a dual-layered solid electrolyte interphase formation on the Li metal electrode and stabilizes the LiNi$_{0.8}$Co$_{0.1}$Mn$_{0.1}$O$_2$-based positive electrode, thus improving interfacial charge-transfer at low temperatures. Consequently, the growth of dendrites at the lithium metal electrode is hindered, thus enabling stable Li||LiNi$_{0.8}$Co$_{0.1}$Mn$_{0.1}$O$_2$ coin and pouch cell operation even at $-30\,°C$. In particular, we report a Li|| LiNi$_{0.8}$Co$_{0.1}$Mn$_{0.1}$O$_2$ coin cell cycled at $-20\,°C$ and 20 mA g$^{-1}$ capable of retaining more than 75% (i.e., around 151 mAh g$^{-1}$) of its first discharge capacity cycle at $30\,°C$ and same specific current.

Owing to better safety, solid-state batteries with polymer electrolytes may replace state-of-the-art Li-ion batteries with flammable organic electrolytes[1–3]. Electro-chemically stable polymer electrolytes appear to be promising for using Li metal anodes (LMAs) and high-nickel layered oxide cathodes (e.g., LiNi$_{0.8}$Co$_{0.1}$Mn$_{0.1}$O$_2$ (NCM811)), thus effectively pushing the cell-specific energy to 450 Wh kg$^{-1}$ and beyond[4–6]. However, batteries with polymer electrolytes (e.g., polyethylene oxide (PEO)-based electrolytes) function reliably at room temperature (e.g., 25–30 °C), but display dramatically reduced energy density, power, and cycling lifetime at temperatures below 0 °C[7–9], which limits battery applications in cold climates.

Poor low-temperature operations are mainly ascribed to insufficient dynamics for the ion transport in the bulk of electrolytes and charge transfer at electrolyte|electrode interfaces, which results in the structural change of solid electrolyte interphase (SEI) interfaces[10,11]. Such issues are magnified for polymer-based electrolytes at low temperatures (e.g., <0 °C)[12]. Introducing organic solvents with low melting points into polymers to form quasi-solid polymer electrolytes can greatly enhance ionic conductivity at low temperatures[13–15]. In addition, conformal electrolyte|electrode interfaces can be formed by in situ polymerization of a non-aqueous liquid precursor, which could accelerate the ionic transport at interfaces[16,17]. However, organic solvents (e.g. ethylene carbonate (EC)) lead to parasitic reactions with

[1]School of Materials Science and Engineering, State Key Laboratory of Material Processing and Die & Mould Technology, Huazhong University of Science and Technology, Wuhan 430074, P. R. China. [2]Laboratory of Advanced Materials and Electron Microscopy, Institute of Physics, Chinese Academy of Science, Beijing 100190, P. R. China. [3]Tianmu Lake Institute of Advanced Energy Storage Technologies Co. Ltd., Liyang, Jiangsu 213300, P. R. China. [4]These authors contributed equally: Zhuo Li, Rui Yu. ✉e-mail: wxf@iphy.ac.cn; xguo@hust.edu.cn

metallic Li, resulting in continuous loss of lithium inventory[18,19], which decreases the cell Coulombic efficiency (CE), shortens the cycling lifetime and causes the dendritic Li plating at the negative electrode[20,21]. Below −15 °C, SEIs derived from organic solvents are highly crystalline and inhomogeneous, with $Li_2CO_3$ being dominant[22,23], while the formation of protective SEI components, such as lithium fluoride nano-salts and amorphous species, are restrained substantially, therefore, the impedance for the interfacial migration of Li ions is increased and the dendrite growth is promoted[24,25]. The use of supramolecular chemistry[26], electrolyte additives[27], polymer molecular engineering[28], and other approaches[29] can improve electrolyte physicochemical properties and stabilize LMAs at low temperatures, however, formed SEIs remain low-ionically conductive and rigid. Consequently, Li metal batteries (LMBs) cannot maintain long-term cycling at low temperatures, and operating temperatures are mostly above −15 °C[28,30,31].

In this work, we report a quasi-solid polymer electrolyte that exhibits a high ionic conductivity of $0.22\,mS\,cm^{-1}$ and a high ionic transference number of 0.8 at −20 °C. Based on the rational design of the interface chemistry, the polymer electrolyte enables a stable and highly conducting dual-layered SEI; consequently, the polymer electrolyte demonstrates good cycling stability toward LMAs at low temperatures. Additionally, the designed electrolyte stabilizes NCM811 cathodes at high voltages (e.g., >4.4 V). Benefiting from these characteristics, operation temperatures of polymer-based batteries are decreased to below −30 °C. During cycling of 200 cycles at −20 °C and $20\,mA\,g^{-1}$, the Li||NCM811 coin cell maintains a high capacity of $>151\,mAh\,g^{-1}$, which is more than 75% of the reversible capacity at 30 °C.

## Results

### Design and preparation of the polymer electrolyte

Li-ion conductivity is a primary concern for electrolytes working at low temperatures. Figure 1a shows the melting point and viscosity of commonly used solvents[32]. By comparison, 2,2,2-Trifluoro-N,N-dimethylacetamide (FDMA) has the lowest melting point of −42 °C, a modest viscosity of 1.1 Cp at 25 °C, which can enhance the ionic transport in polymer electrolytes at low temperatures.

The formation of fluorine-rich interphases contributes to enhancing the interfacial kinetics and stability[33,34]. To control the SEI

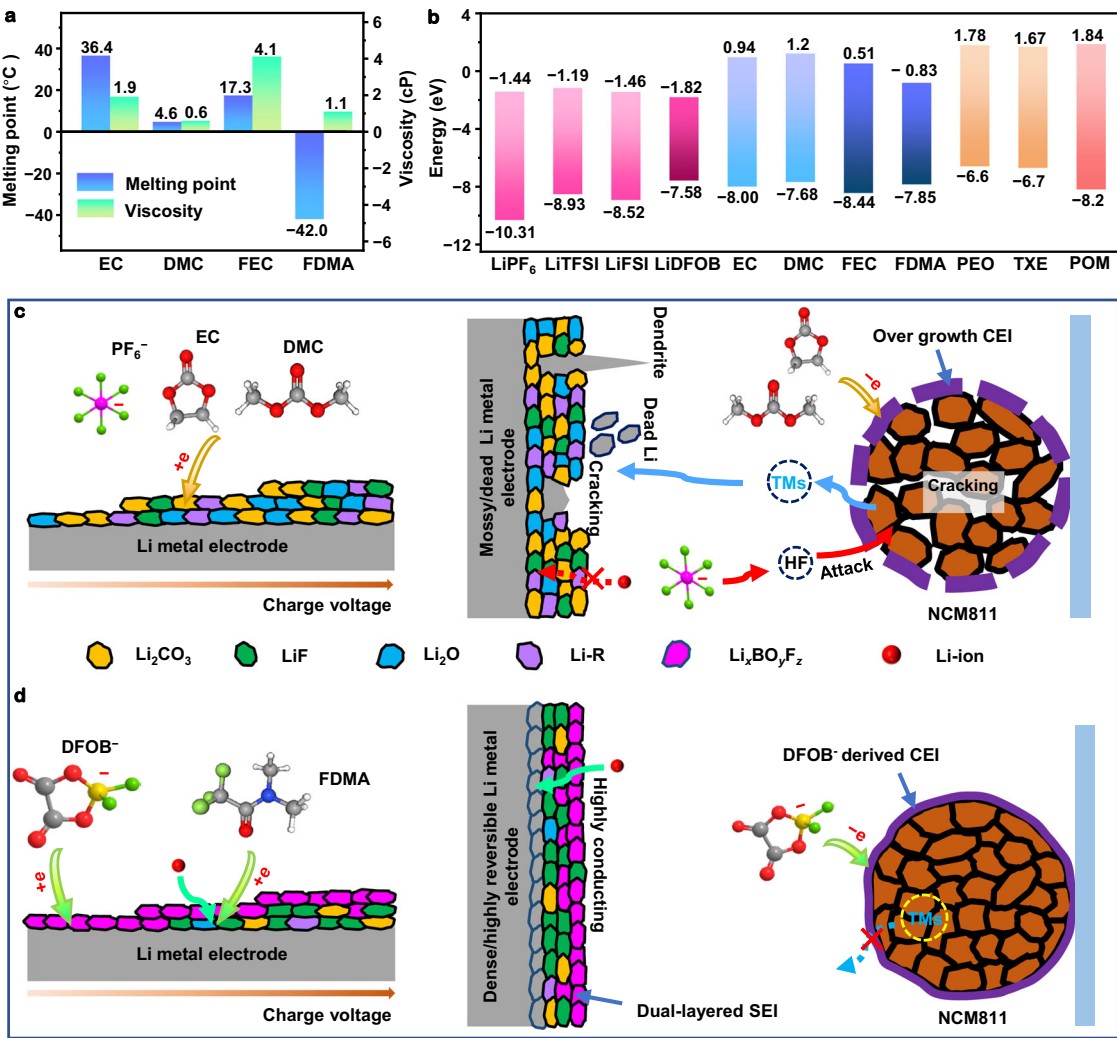

**Fig. 1 | Design of polymer electrolytes for low-temperature Li metal batteries. a** Comparison of the melting points and viscosity (25 °C) of various solvents[32]. **b** HOMO and LUMO energies for commonly used Li salts, solvents, TXE monomer, PEO, and POM polymers. **c** Schematic representation of the solid electrolyte interphase (SEI) formed on the Li metal electrode (left) and degradation processes happening in a Li||NCM811 cell (right) using a non-aqueous carbonate-based electrolyte solution. **d** Schematic representation of the solid electrolyte interphase (SEI) formed on the Li metal electrode (left) and inhibition of the degradation processes in a Li||NCM811 cell (right) using the polymer electrolyte reported in the present research work. The charge voltage refers to the voltage of a Li||NCM811 cell during the charging process.

composition requires that the preferred interface species should have lower LUMO (lowest unoccupied molecular orbital) energies as compared with main components[35]. Meanwhile, a lower HOMO (highest occupied molecular orbital) energy suggests good oxidative stability towards cathodes. Figure 1b and Supplementary Fig. 1 demonstrate the LUMO and HOMO energies of commonly used battery materials, obtained from quantum chemical calculations. Although EC, dimethyl carbonate (DMC) solvents and lithium hexafluorophosphate (LiPF$_6$) salt have larger HOMO−LUMO gaps, organic carbonate electrolytes are incompatible with LMAs, which leads to the overgrowth of solvent-derived SEIs (Li$_2$CO$_3$-rich at low temperatures)[36], resulting in poor interfacial ionic conduction, Li pulverization, low CE, and dendrite growth (Fig. 1c). Additionally, some solvents and LiPF$_6$ salts decompose into organic and inorganic species (C$_2$HO$^-$, ROCO$_2$Li, Li$_2$CO$_3$, HF, etc.) under high voltages (>4.4 V)[37], thereby triggering the generation of a thick cathode|electrolyte interphase (CEI). Then HF passes through the bulky CEI layer and attacks/react with NCM to lead to serious cathode degradations, including cathode−electrolyte side reactions, bulk and surface phase transformation, cracking of the NCM secondary particles, and transition metal (TM) dissolution[4]. Moreover, dissolved TMs eventually waft to the anode side, where they can damage the SEI layer[38].

In contrast, lithium difluoro(oxalato)borate (LiDFOB) exhibits the lowest LUMO, demonstrates a high electron affinity, and can be expected to be reduced firstly, hence it can dominate the formation of the outer-layer SEI. Due to the ionic conduction and electronic insulation, electrons from the anode cannot transport across the Li$_x$BO$_y$F$_z$ layer (formed by the salt decomposition at the negative electrode surface), limiting the reduction of FDMA solvents on the surface[24]. FDMA with strong electron-withdrawing −CF$_3$ and −N− groups shows the second-lowest LUMO energy, therefore, it can contribute to forming a LiF-rich and Li$_2$CO$_3$-less inner-layer SEI under the Li$_x$BO$_y$F$_z$ layer by the acceptance of the electrons from the anode (Fig. 1d)[39,40]. We believe this to be the mechanism, however, further research efforts must be devoted to understanding in detail the whole SEI-formation mechanism. The dual-layered SEI can improve interfacial ionic conduction and chemical/mechanical stability. To increase the oxidative stability of the electrolyte towards cathodes, the utilization of fluoroethylene carbonate (FEC) as a co-polymer with the lowest HOMO level is necessary[38]. Additionally, FEC can regulate the solvation behavior and the SEI formation process, thus stabilizing the electrodes[25]. LiDFOB exhibits a higher HOMO, indicating that it will be the first to be oxidized before solvents form a DFOB-derived passivation layer when the voltage increases, further enhancing the high-voltage stability. Moreover, polyoxymethylene (POM), a polymer of 1,3,5-trioxane (TXE), shows a lower HOMO and a higher LUMO as compared with PEO, thus it can improve the oxidation stability and the reduction stability[41]. In accordance with the above discussion, we prepared a polymer electrolyte by means of in situ polymerization of a TXE-FDMA-FEC-LiDFOB-containing precursor and applied the polymer electrolyte in low-temperature LMBs.

In situ polymerization processes and its mechanism are illustrated in Supplementary information (Supplementary Figs. 2–7 and Supplementary Note 1). The polymer electrolyte was prepared through ring-opening polymerization of a carefully screened precursor, initiated by LiDFOB, which acted as Li salt and initiator (Supplementary Fig. 2)[26]. As confirmed by Fourier transform infrared spectroscopy (FTIR, Supplementary Fig. 3), Raman spectroscopy (Supplementary Fig. 4), nuclear magnetic resonance (NMR, Supplementary Fig. 5), and gel permeation chromatography (GPC, Supplementary Fig. 6 and Supplementary Table 1), the liquid precursor was successfully solidified. Optical photographs in Supplementary Fig. 7 demonstrate the solidification evolution, where the originally flowable liquid precursor turned into a solid-like electrolyte with immovable and stretchy characteristics.

## Electrochemical characteristics of the polymer electrolyte

The ionic conductivity of the polymer electrolyte was measured using electrochemical impedance spectroscopy (EIS) measurements. With the increase of the ratio of FDMA, the ionic conductivity of the electrolyte rapidly increases (Supplementary Fig. 8 and Supplementary Note 2). When the mass ratios of TXE and FDMA are 5:1 and 5:2, the electrolytes have relatively low ionic conductivities at 30 °C. However, at TXE-FDMA mass ratios of 5:5 and 3:5, the electrolytes contain high content of oligomers and solvents (Supplementary Table 1), which not only causes limited thermal/mechanical stability and safety hazards but also deteriorates the electrolyte|electrode interfaces[1]. In view of the balance between the ionic conductivity and the chemical/mechanical/thermal stability, the polymer electrolyte with the TXE-FDMA ratio of 5:3 was chosen as the electrolyte for further investigations. As shown in Fig. 2a and Supplementary Fig. 9, the polymer electrolyte shows an ionic conductivity of 2.5 mS cm$^{-1}$ at 30 °C and an ionic conductivity of 0.22 mS cm$^{-1}$ at −20 °C. Temperature-dependent conductivities of the polymer electrolyte are fitted well using the Arrhenius equation. A low activation energy of 0.33 eV for the ion transport can be derived, indicating a fast Li-ion migration in the polymer electrolyte. The polymer electrolyte also shows a high Li$^+$ transference number of ~0.8 at −20 °C (Fig. 2b), which is comparable to that at 30 °C (0.8) (Supplementary Fig. 10) and higher than those of most polymer-based electrolytes reported in the literature (Supplementary Table 2). The increased Li$^+$ transference number can be attributed to the "assisted Li-ion diffusion" mechanism, in which a lithium-ion is transported from one anion of Li salts to another through binding sites[42,43]. In addition, strong interactions between anions of Li salt and ether oxygen groups in the polymer can also increase the Li$^+$ transference number[44]. The high Li-ion conductivity at low temperatures can alleviate the concentration polarization, thereby improving battery performances[45].

The polymer electrolyte also shows better oxidative and reductive stabilities. In this work, commercially sourced carbonate-based electrolyte (i.e., 1 M LiPF$_6$ in EC:DMC 1;1 v/v non-aqueous liquid electrolyte solution) was used as the reference electrolyte. According to the linear sweep voltammetry (LSV, Supplementary Fig. 11), the reference liquid electrolyte shows a low oxidation potential, as evidenced by a rapid increase in current above ~4.2 V. In contrast, oxidative stability of the polymer electrolyte improves, and no noticeable oxidative current is observed until 5.6 V. Cyclic voltammetry (CV) curves (Supplementary Fig. 12 and Supplementary Note 3) demonstrate that the polymer electrolyte has oxidative or reductive current between −0.5 and 2.5 V, much smaller than those of the reference liquid electrolyte, indicating no side-reaction between the polymer electrolyte and LMA.

## Electrochemical stability towards LMA at low temperatures

Electrochemical characterizations and ex situ postmortem electrode measurements and analyses further validate the stability of the polymer electrolyte towards LMA. Li||Cu coin cells with both the polymer and reference electrolytes were assembled to investigate the Li plating/stripping upon electrochemical cycling at −20 °C. Figure 2c and Supplementary Fig. 13 show CEs evaluated by Li stripping/plating on Cu current collectors; the polymer electrolyte exhibits a significantly enhanced CE and cycling stability over 250 cycles at −20 °C and 0.1 mA cm$^{-2}$, while the CE fades quite fast and the voltage hysteresis increases and fluctuates upon cycling when using the liquid electrolyte under same conditions.

The differences in the cycling stability and the overpotential were also observed in the symmetric Li||Li cells at −20 °C, as shown in Fig. 2d. The thickness of used Li foils is about 50 μm, corresponding to a specific capacity of ~9.65 mAh cm$^{-2}$. Under the measurement condition of 0.2 mAh cm$^{-2}$ (0.1 mA cm$^{-2}$), the voltage curve of the cell using the reference liquid electrolyte shows a rapid increase in polarization after 300 h, and the cell fails after ~500 h, which is caused by unstable SEI

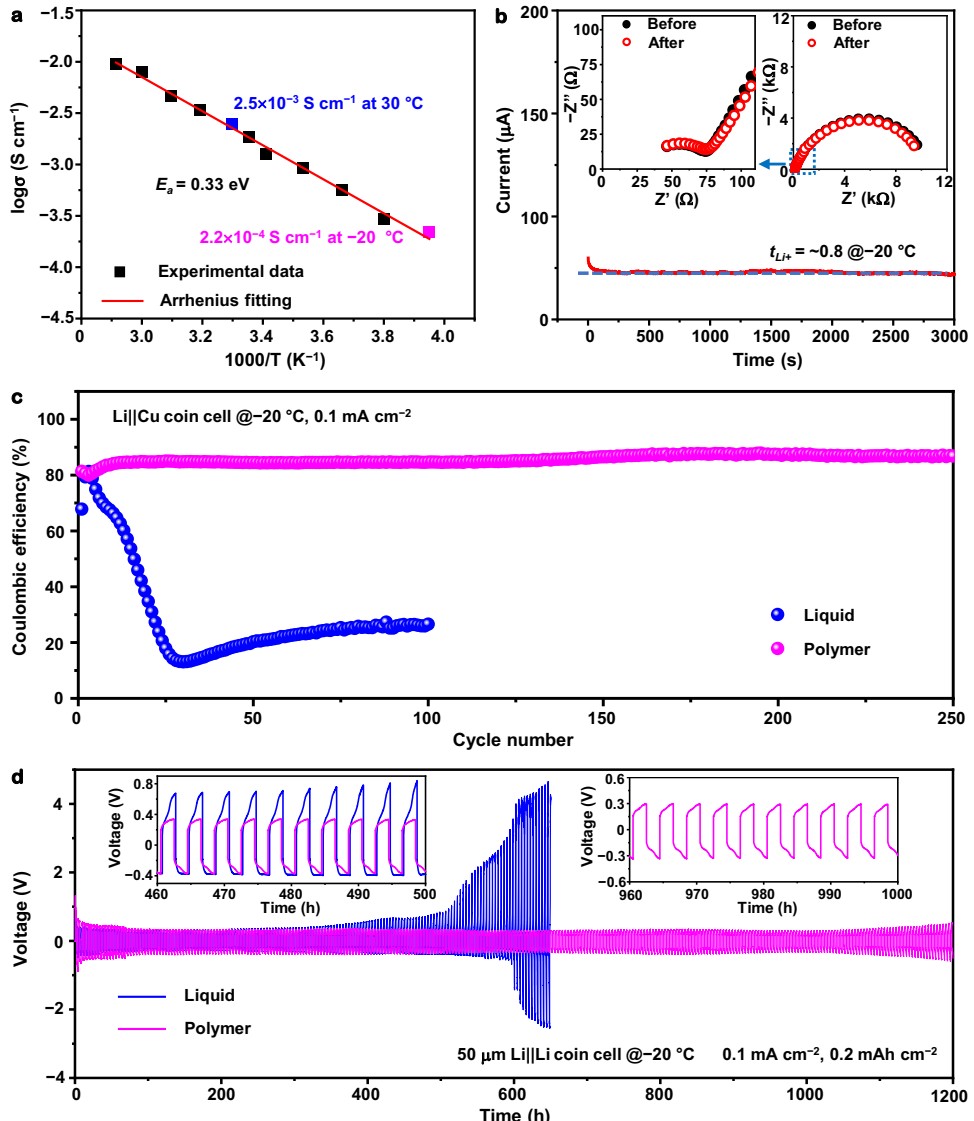

**Fig. 2 | Electrochemical properties of the designed polymer electrolyte at various temperatures. a** Ionic conductivities as a function of temperature. **b** Chronoamperometry profile collected from a symmetric Li‖Li cell (inset shows the EIS plots of the Li‖Li coin cell before and after chronoamperometry). **c** Cell CEs of both electrolytes in Li‖Cu coin cells. **d** Galvanostatic cycling of both electrolytes in symmetric Li‖Li coin cells, insets: the magnification of voltage profiles during 460–500 h and 960–1000 h.

and dendritic Li. In contrast, the cell using the designed polymer electrolyte shows stable cycling for more than 1200 h; and the flat voltage plateau during plating/stripping remains steady throughout the long-term cycling. Even at a higher capacity of 0.4 mAh cm$^{-2}$ (0.2 mA cm$^{-2}$), the Li‖Li cell with the polymer electrolyte still operates for over 250 h (Supplementary Fig. 14). When using a thicker Li of 250 μm, the Li‖Li cell with the polymer electrolyte can be cycled stably over 2500 h at −20 °C and 0.2 mA cm$^{-2}$ (Supplementary Fig. 15). The improved reversibility and cycling stability of the Li plating/stripping may be attributed to the interface chemistry that leads to an SEI having the desired ionic conductivity and chemical/mechanical stability, which is critical for LMBs operating at low temperatures.

## Low-temperature performances of LMBs

Benefiting from the high Li-ion conductivity, the wide electrochemical window, and the good stability towards LMA, the polymer electrolyte can be expected to deliver appealing low-temperature performances in LMBs. A Ni-rich layer cathode NCM811 of commercial interest was selected due to the large capacity (theoretical value of 280 mAh g$^{-1}$);

the combination with a Li anode can result in very high energy density[4,40]. In this context, a 50-μm-thick Li anode and an NCM811 cathode with a mass loading of 2.5 mg cm$^{-2}$ were used. Figure 3a, b and Supplementary Fig. 16 show low-temperature performances of both electrolytes in Li‖NCM811 coin cells. At a specific current of 20 mA g$^{-1}$, the polymer electrolyte enables the Li‖NCM811 coin cell to provide high capacities of 187 mAh g$^{-1}$ at 0 °C and -151 mAh g$^{-1}$ at −20 °C, outperforming the liquid electrolyte system at the same specific current, which only releases capacities of -166 mAh g$^{-1}$ at 0 °C and -130 mAh g$^{-1}$ at −20 °C. Further decreasing the temperature to −30 °C, the coin cell with the reference liquid electrolyte cannot provide any capacity; in contrast, the coin cell with the designed polymer electrolyte can still deliver a modest capacity of -92 mAh g$^{-1}$ at 20 mA g$^{-1}$. The significant capacity-decay of the reference liquid electrolyte in the Li‖NCM811 coin cell blow −20 °C is because the liquid electrolyte is completely solidified, which slows down the charge-transfer kinetics in the electrolyte and through the interfaces.

The Li‖NCM811 coin cell using the designed polymer electrolyte also shows much longer cycling stability at −20 °C. As shown in Fig. 3c

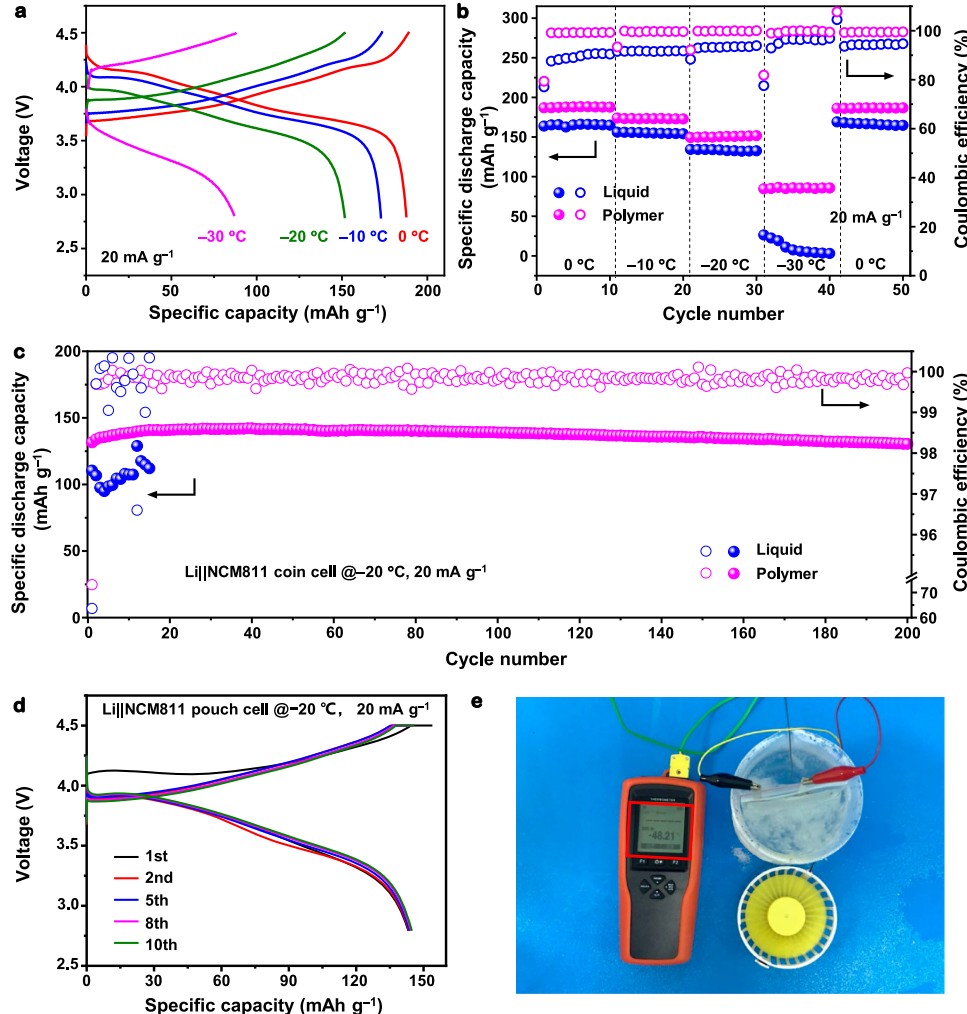

**Fig. 3 | Electrochemical energy storage performances of Li‖NCM811 cells with polymer or liquid electrolytes cycled at various temperatures. a** Charge-discharge voltage profiles (the second cycle at each temperature, 2.8–4.5 V) of the designed polymer electrolyte in the Li‖NCM811 coin cell at different temperatures and 20 mA g⁻¹. **b** Discharge capacities with CEs of both electrolytes in Li‖NCM811 coin cells at different temperatures and 20 mA g⁻¹. **c** Cycling performances of both electrolytes in Li‖NCM811 coin cells at −20 °C and 20 mA g⁻¹. **d** Charge–discharge

voltage profiles of the designed polymer electrolyte in the Li‖NCM811 pouch cell at −20 °C and 20 mA g⁻¹. **e** A Li‖NCM811 pouch cell using the designed polymer electrolyte was powering an electric fan at −48.2 °C. In the above cells, NCM811 cathode with a mass loading of 2.5 mg cm⁻² and anode of 50 μm Li foil was used. The mass of the specific capacity and specific current refers to the mass of the active material in the positive electrode.

and Supplementary Fig. 17, the Li‖NCM811 coin cell using the designed polymer electrolyte achieves a high capacity of nearly 150 mAh g⁻¹ and capacity retention of 99.1% (to the first cycle) with a steady CE of 99.8% over 200 cycles at 20 mA g⁻¹, while the Li‖NCM811 coin cell using the reference liquid electrolyte can only be cycled for 20 cycles and then the capacity and CEs drop rapidly, owing to repeated SEI breaking and reforming. The rate performance of the Li‖NCM811 coin cell with the designed polymer electrolyte at −20 °C is shown in Supplementary Fig. 18; the cell delivers highly reversible specific capacities of 147.3, 122.5, 111.9, 95.1 mAh g⁻¹ at 20, 40, 60, and 100 mA g⁻¹, respectively. When the rate resets to 20 mA g⁻¹, the cell recovers to a capacity of 150.8 mAh g⁻¹, demonstrating good cycling rate performance of the designed polymer electrolyte at low temperatures. Additionally, the designed polymer electrolyte shows good low-temperature performances by pairing with other cathodes, such as LiFePO₄. As shown in Supplementary Fig. 19, the Li‖LiFePO₄ coin cell shows favorable cycling stability and delivers a discharge capacity of ~95 mAh g⁻¹ over 350 cycles at −20 °C and 17 mA g⁻¹.

To further evaluate the application of the designed polymer electrolyte at low temperatures, a single-layer Li‖NCM811 pouch cell (single-side coated Li anode: 50 μm in thickness, and single-side coated

NCM811 cathode: mass loading of 3 mg cm⁻²) was assembled. The pouch cell maintains capacities of ~148 mAh g⁻¹ at −20 °C (Fig. 3d), and ~94 mAh g⁻¹ at −30 °C (Supplementary Fig. 20) for over 10 cycles at 20 mA g⁻¹. Interestingly, the Li‖NCM811 pouch cell could power an electric fan at −48.2 °C in a dry ice/ethanol slush bath (Fig. 3e and Supplementary Movie 1). To the best of our knowledge, it is the lowest working temperature of polymer-based LMBs (as listed in Supplementary Table 2).

**Ex situ postmortem physicochemical characterizations of electrodes**

To gain insights into the SEI formation and microstructural evolution of electrodes, cycled Li anodes (disassembled from Li‖Li coin cells after 100 cycles at −20 °C and 0.2 mA cm⁻²) were collected for postmortem analyses. Scanning electron microscopy (SEM) was employed to investigate the Li deposition morphologies. For the cell using the designed polymer electrolyte, the cycled Li anode shows large granular, uniform, and compact structures (Fig. 4a), and the deposition layer is ~16.8 μm thick (Fig. 4b). In sharp contrast, porous whisker-like Li depositions are observed in the case of the reference liquid electrolyte (Fig. 4c); the active Li is almost completely consumed, and the

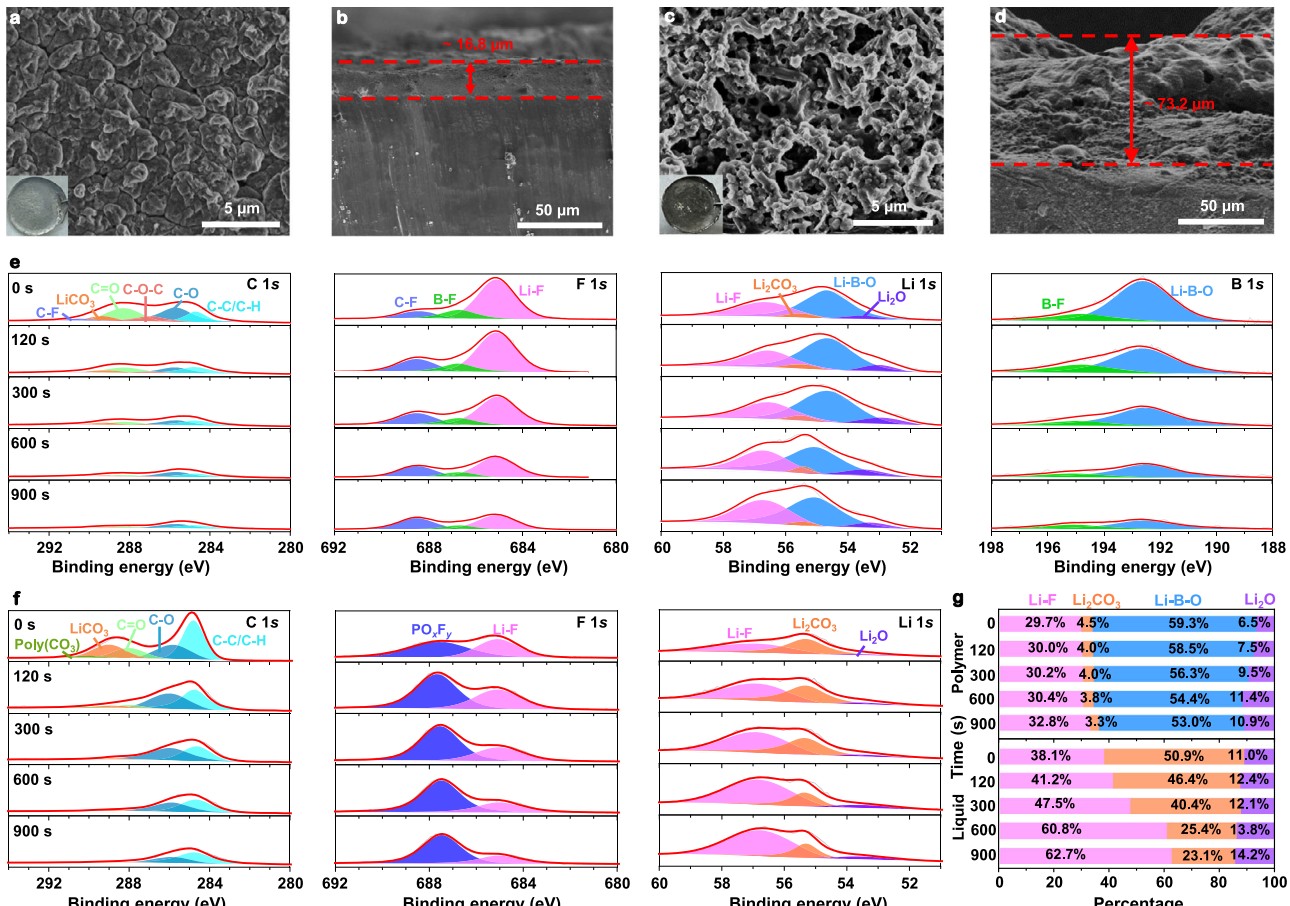

**Fig. 4 | Ex situ postmortem physicochemical characterizations of Li metal electrodes cycled in liquid and polymer electrolytes. a–d** SEM images illustrating morphologies of the LMAs cycled in Li‖Li coin cells with different electrolytes: surface morphologies (**a**) and cross-section views (**b**) using the designed polymer electrolyte, and surface morphologies (**c**) and cross-section views (**d**) using the liquid electrolyte. (insets **a** and **c** show the optical photographs of cycled LMAs). **e**, **f** XPS depth profiles of C 1s, F 1s, B 1s, and Li 1s in LMAs cycled in Li‖Li coin cells with the designed polymer electrolyte (**e**), and the liquid electrolyte (**f**). **g** Relative compositions of Li-containing species. The characterized LMAs were collected from Li‖Li coin cells after 100 cycles at −20 °C and 0.2 mA cm⁻².

thickness of the passivation layer increases to 73.2 μm (Fig. 4d). The less-compact and less-active layer formed by the "dead" Li (i.e., Li metal regions electrically disconnected from the current collector), the thick SEI and the porous morphology, together with the depleted Li inventory and electrolyte, lead to impedance growth and premature cell failure on the anode side[46].

X-ray photoelectron spectroscopy (XPS) depth profiling was conducted to detect the SEI compositions formed in both electrolytes, as shown in Fig. 4e, f. In the C 1s spectra, commonly observed species of C−C/C−H (-284.8 eV), C−O (-285.7 eV), C=O (-288.2 eV), $CO_3^{2-}$ (-289.3 eV) and poly($CO_3$) (-290.3 eV)[47,48] are presented in both electrolytes. However, in the case of the polymer electrolyte, it shows lower $Li_2CO_3$ intensity, and an additional signal of $CF_3$ (290.8 eV)[49] can be detected. The existence of $CF_x$ (-688.4 eV) is further supported by the F 1s spectra[50], two other peaks at -685 and -686.8 eV, corresponding to LiF and B−F[51], respectively, suggest the preferred reactions of FDMA and DFOB⁻ with Li. In the Li 1s spectra of the liquid electrolyte, three components centered at -56.8, -55.4, and -53 eV are clearly detected, which are assigned to LiF, $Li_2CO_3$, and $Li_2O$[22,50], respectively. For the polymer system, one more peak appears in the Li 1s spectrum, centered at -54.6 eV, which is assigned to B-containing species, Li−B−O (corresponding to $Li_xBO_yF_z$)[52], originating from the reduction of DFOB⁻. The $Li_xBO_yF_z$ species (Li−B−O and B−F) can also be observed in the B 1s spectra[51,53].

From the above results, relative compositions of Li-containing species at different depths are collected in Fig. 4g. For both electrolytes, LiF accounts for a large portion of the SEIs. In the case of the reference liquid electrolyte, however, crystalline $Li_2CO_3$ specie becomes the dominant salt on the SEI surface. With increasing depth, the $Li_2CO_3$ content decreases, while the contents of other inorganic species (LiF and $Li_2O$) increase. The highly crystalline SEI with poor ionic conductivity, as well as an inhomogeneous and fragile structure, is unable to accommodate the large interfacial fluctuations and morphological changes so that they could be damaged by the interfacial stress/strain caused by the Li plating/stripping[54], which results in loss of the lithium inventory. In contrast, for the polymer system, $Li_xBO_yF_z$ and LiF species dominate the SEI chemistry. With increasing sputtering depth, the $Li_xBO_yF_z$ content decreases while the LiF species increase. Therefore, an outer layer of $Li_xBO_yF_z$ and an inner layer of LiF can largely account for the SEI in the polymer system.

The time-of-flight secondary ion mass spectroscopy (ToF-SIMS) also provided supporting information for the dual-layered SEI. As shown in Fig. 5a, less $C_2HO^-$ (representing organic species)[37] and crystalline $CO_3^{2-}$ (representing $Li_2CO_3$ species) species are found on the SEI surface when using the designed polymer electrolyte. Particularly, $BOF_2^-$-containing species (representing $Li_xBO_yF_z$)[29] uniformly cover the Li anode and dominates the outer layer ingredient of the SEI; while the amount of $LiF_2^-$ (representing LiF) species are present in the inner phase of the SEI. A similar study was conducted on the SEI formed in the liquid electrolyte; the outer layer of the SEI is abundant with $C_2HO^-$ and $CO_3^{2-}$ species (Fig. 5b), suggesting continuing reactions between the Li anode and carbonate solvents. Figure 5c, d shows

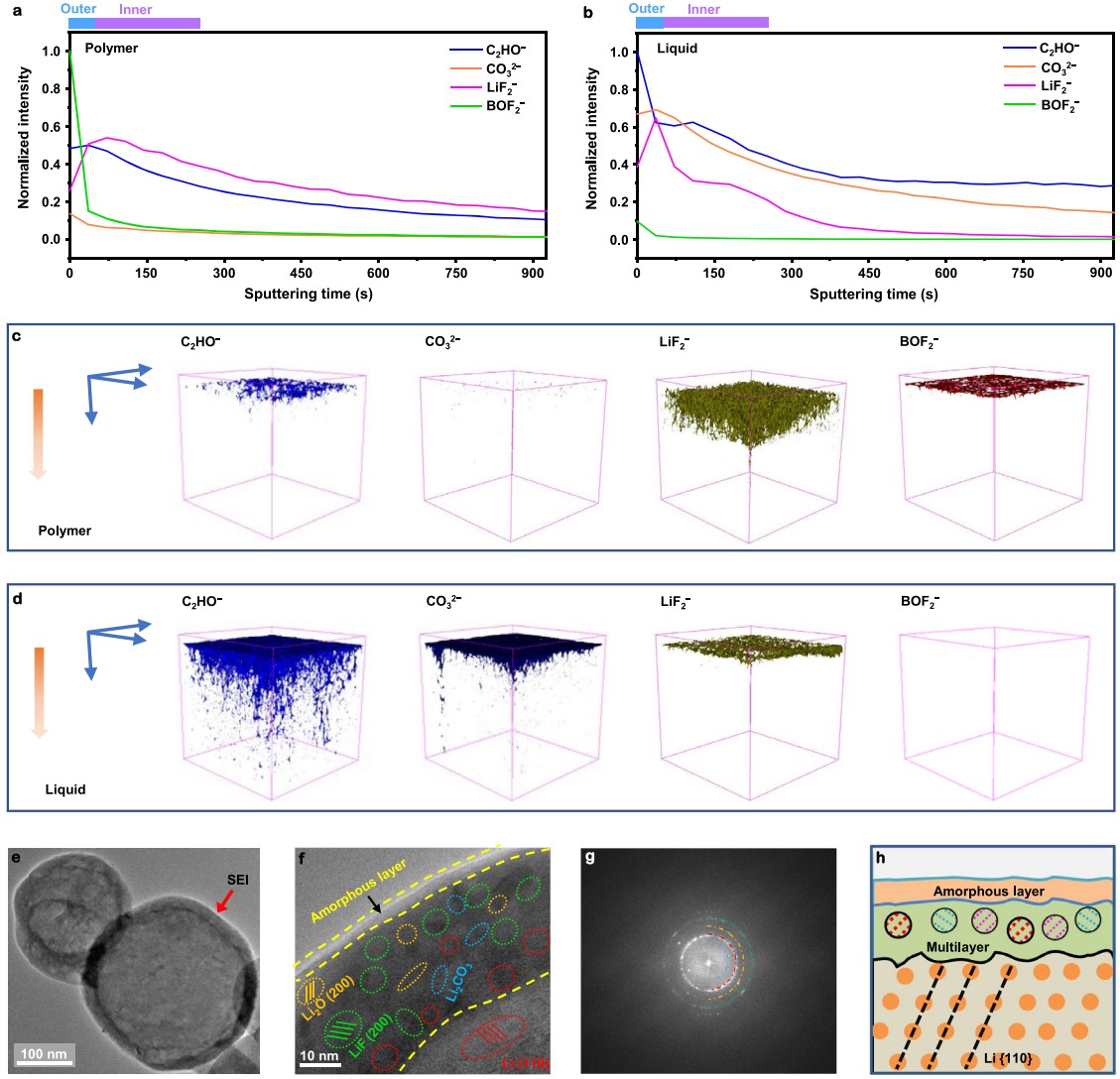

**Fig. 5 | Physicochemical characterizations of the Li metal electrode SEI via ex situ postmortem measurements. a, b** Depth profiles of the Li SEIs in Li||Li coin cells with the designed polymer electrolyte (**a**), and the liquid electrolyte (**b**). **c, d** 3D renders of the Li SEIs in Li||Li coin cells with the polymer electrolyte (**c**), and the liquid electrolyte (**d**). The characterized LMAs were collected from Li||Li coin cells after 100 cycles at −20 °C and 0.2 mA cm⁻². **e, f** Cryo-TEM images of deposited Li in the cell with the polymer electrolyte at different scales. **g** Corresponding FFT pattern of the inner SEI: green circle: LiF; red circle: Li; yellow circle: $Li_2O$; blue circle: $Li_2CO_3$. **h** Schematic of the observed dual-layered SEI on the deposited Li. The Li sample for cryo-TEM was prepared by depositing Li on a TEM grid, at −20 °C and 0.2 mA cm⁻² cm (to a capacity of 0.02 mAh cm⁻²), with the designed polymer electrolyte.

the 3D rendering of the ToF-SIMS top-down depth sputtering, clearly demonstrating that a dual-layered SEI with a $Li_xBO_yF_z$ dominating outer layer and a LiF-rich inner layer fully covers the Li anode in the polymer system (Fig. 5c). The dual-layered SEI derived from the designed polymer electrolyte is different from conventional low-temperature-formed SEIs (Fig. 5d) that display a highly crystalline and $Li_2CO_3$-dominant structure.

To further reveal the dual-layered structure, we studied the nanostructure of the low-temperature-formed SEI by using cryo-transmission electron microscopy (cryo-TEM). In general, a continuous and uniform SEI is found on the surface of the deposited Li, as shown in Fig. 5e. When the magnification is increased to the atomic scale, a dual-layered SEI with inorganic inner-phase and the amorphous outer layer is observed in the polymer system (Fig. 5f). The inner inorganic-rich layer is comprised of small amounts of $Li_2CO_3$ and $Li_2O$ species and large amount of LiF, being consistent with the XPS and ToF-SIMS results. Characteristic bright diffraction spots corresponding to the LiF (200) plane are captured in the fast Fourier transform (FFT) pattern (Fig. 5g). The schematic of the observed SEI on Li is

shown in Fig. 5h. The dual-layered SEI suppresses the Li–electrolyte interactions, minimizes loss of the lithium inventory, and improve the cycling performance of Li metal electrodes.

In addition to the anode side, the designed polymer electrolyte stabilizes NCM811 cathodes at low temperatures. Intergranular cracking between connected primary particles is a critical issue for the degradation of Ni-rich cathodes. The surface phase transition from a layered to resistive rock-salt NiO-like structure is known to degrade the NCM cathode performance. To characterize such microstructural degradation, the NCM811 cathodes (disassembled from the Li||NCM811 coin cell after 100 cycles at −20 °C and 40 mA g⁻¹) were sampled and ex situ investigated (Fig. 6a–h). SEM images in Fig. 6a show that the cycled NCM811 cathode in the polymer electrolyte system maintains a spherical and smooth surface, and the extensive intergranular cracking is apparently suppressed or delayed by using the designed polymer electrolyte. The origin of these beneficial effects could be attributed to the formation of a thin and uniform CEI layer (~2 nm), as outlined by the yellow lines in the TEM image (Fig. 6c). In comparison, structural damage of the cycled NCM811 cathode in the liquid system is visible

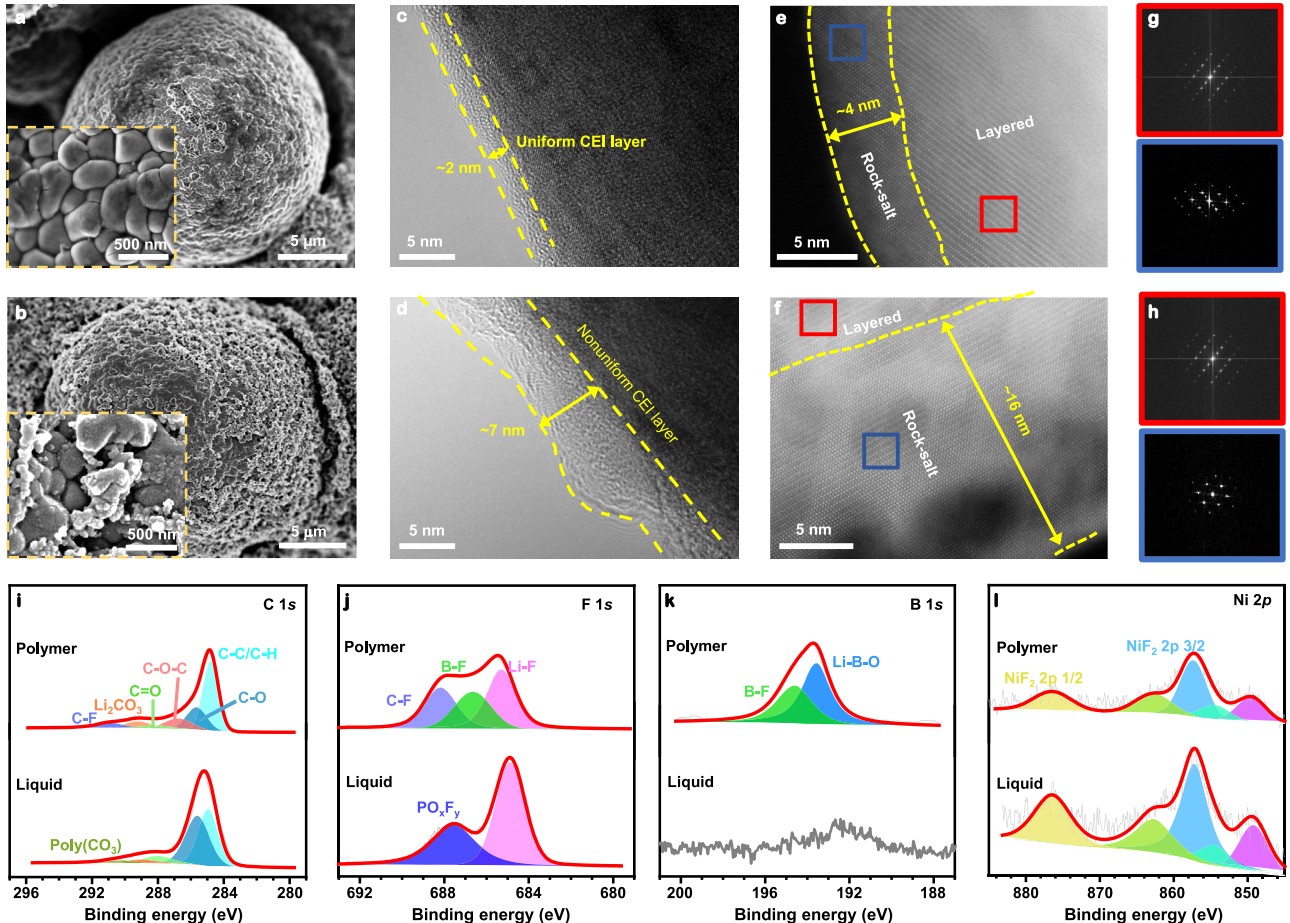

**Fig. 6 | Ex situ postmortem physicochemical characterizations of NCM811-based electrodes cycled in liquid and polymer electrolytes. a, b** SEM images of NCM811 particles using the designed polymer electrolyte (**a**), and the liquid electrolyte (**b**). **c, d** TEM of NCM811 particles using the polymer electrolyte (**c**), and the liquid electrolyte (**d**). **e–h** High-resolution TEM images and corresponding FFT for NCM811 particles cycled with the polymer electrolyte (**e, g**), and the liquid electrolyte (**f, h**). **i–l** XPS spectra of C 1*s* (**i**), F 1*s* (**j**), B 1*s* (**k**), and Ni 2*p* (**l**) for NCM811 cathodes using both electrolytes. The NCM811-based electrodes were collected from Li‖NCM811 coin cells after 100 cycles at −20 °C and 40 mA g⁻¹. The specific current refers to the mass of the active material in the positive electrode.

(Fig. 6b), resulting in more electrolyte-exposing fresh surfaces and extensive CEI formation; therefore, a much thicker (~7 nm) and uneven CEI can be observed, as confirmed by the TEM image in Fig. 6d. Furthermore, for the liquid electrolyte, the surface undergoes a phase transition, with an accumulation of ~16 nm disordered rock-salt phase on the cathode surface (Fig. 6f), caused by electrolyte-NCM811 interactions. The surface layers of the NCM811-based electrode are subject to a certain degree of intermixing, where Li sites are partially occupied by anti-site Ni ions. Such reconstruction also exists in the NCM811 electrode cycled in the polymer electrolyte system, with a rock-salt thickness of ~4 nm (Fig. 6e). The phase transition of the NCM811 cathodes is also confirmed by the FFT patterns shown in Fig. 6g, h, revealing the presence of a rock-salt phase (blue zone) and a layered phase (red zone).

To further characterize the chemical states and components of the CEIs, XPS investigations were conducted on the surfaces of the cathodes after 100 cycles at −20 °C and 40 mA g⁻¹ (Fig. 6i–l). Compared with the cathode cycled in the reference liquid electrolyte, the one cycled in the designed polymer electrolyte has a much weaker C 1*s* signal (Fig. 6i, especially the peaks that can be attributed to organic species), F 1*s* signal (Fig. 6j, especially the peak that can be attributed to F–Li, B–F, and P–O–F), and Ni 2*p* signal (Fig. 6l, contributed by Ni-containing CEI components, such as NiF₂)[55], indicating the suppressed CEI growth and TM dissolution. Meanwhile, strong Li–B–O and B–F signals are observed in the polymer system (Fig. 6k), indicative of

Li$_x$BO$_y$F$_z$-rich inorganic components[56]. Therefore, the CEI derived from the designed polymer electrolyte should consist of more Li$_x$BO$_y$F$_z$ components, which are known to be 'good' CEI components, so that the CEI can dynamically inhibit side reactions, maintain the initial structure of NCM and reinforce the interfacial stability[52,56].

## Li metal cell testing at 30 °C

The designed polymer electrolyte also enables highly reversible Li-metal stripping/plating (Li CE of >98%), and compact morphology, thus minimizing the Li loss and volumetric expansion at 30 °C (Supplementary Figs. 21–23 and Supplementary Note 4). Therefore, the designed polymer electrolyte in the Li‖NCM811 coin cell should improve the battery performance also at 30 °C. As shown in Fig. 7a, b and Supplementary Fig. 24, the discharge capacities are almost identical for both electrolytes from the specific currents of 40–100 mA g⁻¹. When the charge current is higher than 200 mA g⁻¹, a lower capacity is found for the liquid system, and a very limited capacity is obtained at rates above 1000 mA g⁻¹. These delays may be caused by damage to the Li anode and the NCM cathode at high rates during the fast charge–discharge. In comparison, the designed polymer electrolyte performs better at high rates, and the cell shows stable capacities of ~198 mAh g⁻¹ at 40 mA g⁻¹, ~182 mAh g⁻¹ at 200 mA g⁻¹, ~164 mAh g⁻¹ at 600 mA g⁻¹, ~150 mAh g⁻¹ at 1000 mA g⁻¹, and ~118 mAh g⁻¹ at 2000 mA g⁻¹. The specific capacity then recovers to ~197.6 mAh g⁻¹ after the rate is decreased back to 40 mA g⁻¹. These

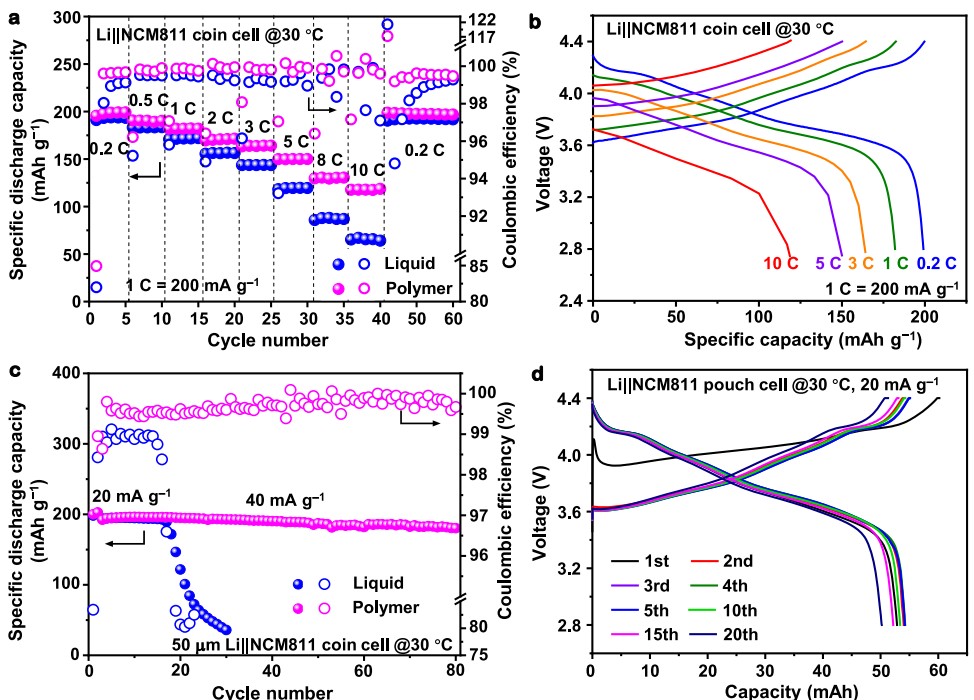

**Fig. 7 | Electrochemical energy storage performances of Li‖NCM811 cells with polymer or liquid electrolytes cycled at 30 °C. a** Rate performances of both electrolytes, and **b** charge–discharge voltage profiles (second cycle) of the Li‖NCM811 coin cell with the designed polymer electrolyte at different applied currents. In these coin cells, NCM811 cathode with a mass loading of 2.5 mg cm$^{-2}$ and anode of 50 µm Li foil was used. **c** Cycling performance of Li‖NCM811 coin cells with both electrolytes under practical conditions. The first two formation cycles were carried out at 20 mA g$^{-1}$, and the long-term cycling was set at 40 mA g$^{-1}$. **d** Charge–discharge voltage profiles of the Li‖NCM811 pouch cell with the polymer electrolyte. In the Li‖NCM811 coin and pouch cell under practical conditions, high-loading NCM811 cathodes (~2.5 mAh cm$^{-2}$) and 50 µm Li foil anodes were used, where the N/P ratio was ~3.86 and the E/C ratio was ~5 g (Ah)$^{-1}$. The mass of the specific capacity and specific current refers to the mass of the active material in the positive electrode.

results clearly show that the designed polymer electrolyte enables fast charge-discharge processes in LMBs. Furthermore, after two formation cycles (20 mA g$^{-1}$), the designed polymer electrolyte also shows better cycling performances at 30 °C and 100 mA g$^{-1}$, delivering an initial capacity of ~195 mAh g$^{-1}$, and 96.7% capacity retention with an average CE of 99.7% over 200 cycles (Supplementary Fig. 25).

In practical applications of LMBs, a high-loading cathode, lean electrolyte, and limited Li anode are simultaneously required. As shown in Fig. 7c and Supplementary Fig. 26, even applying relevant practical conditions (cathode loading of ~2.5 mAh cm$^{-2}$, negative to positive (N/P) ratio of ~3.86, electrolyte to capacity (E/C) ratio of ~5 g (Ah)$^{-1}$), the Li‖NCM811 coin cell with the designed polymer electrolyte enables improved cycling stability, achieving an initial capacity of 196 mAh g$^{-1}$ and 94% capacity retention over 80 cycles at 40 mA g$^{-1}$. When using the reference liquid electrolyte, the cell under the same condition survives only 20 cycles, because uncontrolled side reactions rapidly deplete the Li and/or electrolyte, causing capacity decay. Furthermore, a single-layer Li‖NCM811 pouch cell with single-side coated electrodes (N/P ratio is ~3.86, E/C is ~5 g (Ah)$^{-1}$) maintains a practical discharge capacity of ~54.5 mAh over 20 cycles at 20 mA g$^{-1}$ (Fig. 7d). Notably, the cell could power an electric fan under normal, folding, and corner-cut conditions (Supplementary Fig. 27).

## Discussion

The dual-layered SEI formed on the Li metal electrode, consisting of an amorphous Li$_x$BO$_y$F$_z$ outer layer and a LiF-rich inner layer, provides synergistic effects: (1) Li$_x$BO$_y$F$_z$ and LiF are good electronic insulators, each with a large electrochemical window, therefore, the SEI restrains the electrolyte decomposition and the dendrite formation[33]; (2) the Li$_x$BO$_y$F$_z$/LiF-rich and Li$_2$CO$_3$-poor SEI has a low energy barrier for the Li$^+$ diffusion, facilitating the Li$^+$ transfer across the interfaces and

promoting uniform deposition of Li[57]; (3) the formation process of Li$_x$BO$_y$F$_z$ excludes high-temperature calcination so that the products are inclined to deliver amorphous state and thus lead to high-plasticity properties, which can mechanically accommodate the volume change of the electrode[52]. These properties assist to suppress lithium dendrite formation, promote stable Li deposition and facilitate interfacial conducting, thus achieving good battery performances at low temperatures.

In summary, we demonstrate a polymer-based electrolyte synthesized via in situ polymerization of a precursor containing TXE-FDMA-LiDFOB; the electrolyte enables fast ion transport and reversible cycling at low temperatures. The rationale behind this is the relatively low LUMO levels of FDMA and LiDFOB, which initiates decomposition into a dual-layered SEI with less Li$_2$CO$_3$ phase, distinctively different from the crystalline structure of conventional low-temperature SEIs. Meanwhile, the designed polymer electrolyte effectively stabilizes the NCM811 cathode by in situ building an amorphous CEI, thus suppressing side reactions, phase transformation, and stress-corrosion cracking. Under the protection of both SEI and CEI layers, the formation of Li dendrites and "dead" Li are suppressed, and degradations of both electrodes are effectively prevented. Consequently, operation temperatures of the polymer-based Li‖NCM811 pouch cells are decreased to −48.2 °C, and the Li‖NCM811 coin cell maintains stable cycling over 200 cycles at 100 mA g$^{-1}$, delivers high initial capacities of ~198 mAh g$^{-1}$ at 30 °C (100 mA g$^{-1}$), 151 mAh g$^{-1}$ at −20 °C (20 mA g$^{-1}$), and 92 mAh g$^{-1}$ at −30 °C (20 mA g$^{-1}$).

## Methods
### Materials
Active materials (LiNi$_{0.8}$Co$_{0.1}$Mn$_{0.1}$O$_2$ (NCM811) or LiFePO$_4$ powders), Super C65, polyvinylidene fluoride (PVDF), anhydrous N-methyl-2-

pyrrolidone (NMP), carbon-coated aluminum (Al, 15 μm-thick) foil and $Al_2O_3$-coated polyethene (PE, 16 μm thick, 43% porosity) were purchased from the Guangdong Canrd New Energy Technology Co., Ltd. 1,3,5-trioxane (TXE, >99.0% (GC)) and 2,2,2-trifluoro-N, N-dimethylacetamide (FDMA, >98.0% (GC)) were purchased from Tokyo Chemical Industry Co., Ltd. FEC (99.95%), DMC (99.95%), LiDFOB (99.9%) and the reference carbonate electrolyte (i.e., 1M $LiPF_6$ in EC:DMC 1;1 v/v non-aqueous liquid electrolyte solution) were purchased from DoDoChem. Li foils (50 or 250 μm) were purchased from China Energy Lithium Co., Ltd. FDMA was dried with 4 Å molecular sieves (Sigma–Aldrich) before use.

## Positive electrode preparation

The positive electrolyte laminates (NCM or LFP) were prepared by a slurry-coating method without calendering step in an Ar glovebox ($H_2O$ < 0.1 ppm, $O_2$ < 0.1 ppm). Typically, 85 wt% active materials, 10 wt% Super C65 (as conductive carbon), and 5 wt% PVDF (as binder), were wetly mixed together in the NMP solvent (5 wt%) for 12 h by using a stirrer. And then, the slurry mixture was cast onto a carbon-coated Al foil and then dried at 80 °C for 12 h in a vacuum. The mass loading of active materials on the positive electrode laminates was around 2.5 or 12.5 mg cm$^{-2}$.

## Polymer electrolyte preparation

Electrolytes were prepared in an argon gas-filled glovebox ($H_2O$ < 0.1 ppm, $O_2$ < 0.1 ppm). TXE monomer, FDMA, and FEC solvent were added in a polypropylene container at a mass ratio of 5:3:1. The mixture was melted and stirred at 60 °C for 10 min by using a magnetic stirrer. Then, 1 M LiDFOB as the lithium salt and initiator was then completely dissolved in the mixture to obtain the precursor solution. Subsequently, the precursor was immediately injected into a coin or pouch Li metal cell, in which the $Al_2O_3$-coated PE was used as the separator, and then assembled cells were kept at 80 °C for 2 h to realize spontaneous polymerization, therefore, quasi-solid-state Li metal cells were obtained.

## Physicochemical characterizations

The precursor solution and polymer electrolyte were dissolved in Chloroform-d1 for $^1$H NMR and $^{13}$C NMR analysis on a Bruker AVANCE III 600 MHz. GPC investigations were performed by dissolving the polymer in THF and eluting the solutions in an ambient temperature GPC (Waters) equipped with triple detection capability for absolute polymer molecular weight determination. Fourier transform infrared spectra of the precursor solution and the polymer electrolyte were characterized in the attenuated total reflection mode (Thermo Scientific, Nicolet iS50R). Raman spectroscopy investigations were conducted by a Micro-laser confocal Raman spectrometer (Horiba LabRAM HR800, France) at 30 °C.

For postmortem analyses, Li||Li coin cells (after 100 cycles at −20 °C and 0.2 mA cm$^{-2}$) and Li||NCM811 coin cells (after 100 cycles at −20 °C and 40 mA g$^{-1}$) were disassembled to collect the Li foils or the NCM cathodes. Subsequently, these samples were washed with DMC solvent three times inside a glovebox ($H_2O$ < 0.1 ppm, $O_2$ < 0.1 ppm), and then sealed in airtight containers in the glovebox before being transferred for further characterization. SEM (Nova NanoSEM 450) was used to observe the surface morphology of cycled electrodes. The chemical compositions of SEIs and CEIs were studied by XPS (PHI 5000 VersaProbe III). Argon ion sputtering was adopted for the depth profile analysis on the Li electrode. The element distributions at different depths in the cycled Li anode were analyzed using the ToF-SIMS (PHI nano TOF II). Cryo-TEM (JEM-F200) was used to analyze the composition and structure of SEIs. Cryo-TEM samples were prepared by electrochemical deposition of Li onto 400-mesh Cu grids with lacey carbon films for 10 min in a coin cell. The Li sample for cryo-TEM was prepared by depositing Li on the TEM grid, at −20 °C and 0.2 mA cm$^{-2}$ (to a capacity of 0.02 mAh cm$^{-2}$), with the designed polymer

electrolyte. The sample was then dabbed with DME and frozen at −175 °C on the cryo-transfer holder (Fischione 2550), to carry out the cryo-TEM according. TEM samples were harvested from the disassembled NCM811 electrodes and a secondary particle of NCM811 was selected randomly to characterize the morphology and microstructure by using TEM (JEM-ARM200F).

## Electrochemical measurements and battery testing

The electrochemical measurements and battery testing were carried out in an environmental chamber. LSV (2.5–6 V) and CV (−0.5 to 2.5 V) tests were conducted with Li||Pt coin cells at a scanning rate of 2 mV s$^{-1}$ and 30 °C via an electrochemical workstation (Gamry, Interface-1000E). Ionic conductivities were measured by EIS using a homemade glass bottle (20 mm in diameter) containing the polymer electrolyte with two pieces of stainless steel inserted inside the electrolyte (Supplementary Fig. 28); the testing temperatures ranged from −20 to 60 °C, the amplitude voltage was 10 mV, and the frequency range was from 5 MHz to 1 Hz. The ionic conductivities ($\sigma$) of the polymer electrolyte were calculated according to the equation:

$$\sigma = d/SR \quad (1)$$

where $d$ (cm) is the thickness of the polymer electrolyte, $S$ (cm$^2$) is the effective contact area with the stainless steel, and $R$ (Ω) is measured by EIS. The Li$^+$ transference number was determined at −20 and 30 °C by combining EIS and chronoamperometry with a DC polarization voltage ($\Delta V$) of 10 mV using Li||Li coin cells. The Li$^+$ transference number ($t_{Li^+}$) was calculated by the following equation[58]:

$$t_{Li^+} = \frac{I^s R_b^s (\Delta V - I^0 R_i^0)}{I^0 R_b^0 (\Delta V - I^s R_i^s)} \quad (2)$$

where $I^0$ and $I^s$ are the initial and steady currents during polarization, $R_b^0$ and $R_b^s$ are the initial and steady bulk resistances, $R_i^0$ and $R_i^s$ are the initial and steady interfacial resistances.

To study the electrochemical stability between Li and electrolytes, Li||Li symmetric cells and Li||Cu coin cells were assembled and tested at −20 and 30 °C by using a LAND battery testing system (CT3001A, Wuhan LANHE Electronics Co., Ltd.). The galvanostatic charge/discharge tests of Li||NCM811 and Li||LiFePO$_4$ coin or pouch cells were performed to evaluate the cycling performance and rate capabilities. The mass of the applied specific current and specific capacity referred to the mass of the active material in the positive electrode. All cells (coin and pouch types) were tested in an environmental test chamber to maintain a stable temperature environment (−60 to 150 °C, MT3065, Guangzhou-GWS Environmental Equipment Co., Ltd.).

In the batteries working under practical conditions, $N/P = Q_A/Q_C = Q_{Li}m_{Li}/Q_C$, where $Q_A$ is the area capacity of the anode, and $Q_{Li}$ is the theoretical specific capacity of lithium metal, which is 3860 mAh g$^{-1}$, $m_{Li}$ is the weight of the Li metal. For the 50 μm-thick Li anode, $m_{Li}$ is ~2.5 × 10$^{-3}$ g cm$^{-2}$, and the area capacity can be calculated to be ~9.65 mAh cm$^{-2}$; then the N/P ratio of our battery is ~3.86. Single-layer Li||NCM811 pouch cells were assembled with the designed polymer electrolyte (capacity: ~60 mAh) in an Ar glovebox ($H_2O$ < 0.1 ppm, $O_2$ < 0.1 ppm). Of which a single-side coated NCM811-based positive electrode (4.4 × 5.7 cm$^2$, 12.5 mg cm$^{-2}$, 2.5 mAh cm$^{-2}$), single-layer Li metal negative electrode (4.5 × 5.8 cm$^2$, 50-μm-thick, 9.65 mAh cm$^{-2}$) and a $Al_2O_3$-coated PE separator (6 cm in width, 16 μm in thickness) between the cathode and anode were composed.

## Quantum chemical calculations

Quantum chemical calculations of HOMO and LUMO energies were conducted using density functional theory with the Gaussian 16 software package with a basis set of B3LYP functional[59] and 6–31 G (d)[60].

## Reporting summary

Further information on research design is available in the Nature Portfolio Reporting Summary linked to this article.

## Data availability

The authors declare that all experimental data and relevant analysis of this work are available from the corresponding author upon reasonable request.

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

## Acknowledgements

The authors would like to express their appreciation to the National Natural Science Foundation of China and the Israeli Science Foundation for funding this research within the framework of the joint NSFC-ISF (51961145302). We acknowledge the funding from the National Key Research and Development Program of China (Grant no. 2022YFB 2502200) and the Natural Science Foundation of Beijing (Grant No. Z200013). This work was also supported by the China Postdoctoral Science Foundation-funded project (2020M682403). The analytical and testing center of Huazhong University of Science and Technology was acknowledged for the FTIR, SEM, and XPS investigations.

## Author contributions

Z.L. and R.Y. performed the experiments, S.W. and X.W. conducted cryo-TEM investigations, and Z.L., R.Y., and Q.Z. carried out data analyses. Z.L., R.Y., X.W., and X.G. wrote the paper. X.G. supervised the work. Z.L. and R.Y. contributed equally to this work.

## Competing interests

The authors declare no competing interests.
