## [Peer Review File · Nature Communications]

REVIEWER COMMENTS

Reviewer #1 (Remarks to the Author):

This work reports a novel polymer electrolyte with high ionic conductivity and ionic transference number at low temperature. The Li||NCM811 battery assembled with such the polymer electrolyte presented high capabilities and stable long-term cycling at both room and low temperatures ($-20\text{ }^{\circ}\text{C}$) due to good electrochemical stability and fast kinetics. Moreover, the operation temperature of the batteries can be decreased to below $-30\text{ }^{\circ}\text{C}$, and even at $-58.3\text{ }^{\circ}\text{C}$. These results are very important, which can effectively promote the safety batteries to work at ultra-low temperatures. Therefore, this manuscript should be published in Nature Communications after addressing the following issues:

1. The authors briefly mentioned that FDMA with low melting point and a modest viscosity can greatly enhance the ionic transport in the electrolyte. The ionic conduction mode in the polymer electrolyte should be clarified.
2. At both low and room temperatures, the polymer electrolyte shows significantly enhanced Li^+ transference number, which means that the transportation of anions is selectively restricted. The authors had better provide a more detailed explanation.
3. From the SEM images, it can be found that both the thicknesses of the Li deposition layer and the chunk size of Li on the surface clearly became smaller from room temperature to $-20\text{ }^{\circ}\text{C}$. Can the authors explain this phenomenon?
4. It is not clear whether the measured ionic conductivity is due to the polymer electrolyte with a PE separator or only the polymer electrolyte. The authors should give details of the measurement method.
5. The authors need to explain how to calculate the N/P ratio.
6. This manuscript only gives the LSV curve and CV curves of the polymer electrolyte. I recommend the authors provide these data of the reference electrolyte as well.
7. Some electrochemical tests were performed under different conditions with respect to cathode loading, temperature, C-rate, etc., which might be confusing for readers. Some data, such as cathode loading, temperature, and C-rate, on each figure, should be presented.
8. The authors suggested that amorphous $\text{Li}_x\text{BO}_y\text{F}_z$ can mechanically accommodate the large volume change of the electrode; however, detailed explanation is lack. Can the authors elaborate?

Reviewer #2 (Remarks to the Author):

The authors present a polymer-based electrolyte which strategically influences the composition of the SEI to promote low temperature operation of a lithium ion battery utilizing a lithium metal anode. This manuscript is of interest to the scientific community and suggested for publication after a number of major revisions are addressed.

- 1) Please define all acronyms upon first introduction in the manuscript

2) The polymer electrolyte described herein is most likely a gel polymer electrolyte. Please describe the electrolyte as a gel polymer electrolyte within the manuscript or provide evidence that there are no liquid electrolyte domains within a solid polymer matrix in the electrolyte described in this manuscript.

3) The following statement in the introduction requires correction or clarification:
"In contrast, LiDFOB exhibits the lowest LUMO, demonstrates a high electron affinity and can be expected to be reduced firstly, hence it can dominate the formation of the outer-layer SEI; while FDMA with strong electron-withdrawing $-CF_3$ and $-N-$ groups shows the second-lowest LUMO energy, it can contribute to form LiF-rich and Li_2CO_3 -less inner-layer SEI (Fig. 1d)."

If LiDFOB were reduced first, this would contribute to the inner-layer SEI and FDMA would contribute to the outer layer SEI.

4) Remove unnecessary articles such as "the" in "the Raman spectroscopy" and "the nuclear magnetic resonance" and "the mass spectrometry"

5) The operation of the pouch cell at -58.3 C needs to be conducted in a thermally equilibrated temperature bath. The use of dry ice in its solid state provides insufficient thermal contact to provide confidence in the overall pouch cell temperature. The use of a slush bath is advised to provide adequate thermal contact and confidence in the equilibrated temperature of the pouch cell. See DOI: 10.1002/adv.202106032 for an example of this.

6) Please highlight the equipment used to ensure temperature stability during cycling studies in the experimental section.

Reviewer #3 (Remarks to the Author):

article id:374713

This work is interesting and timely. The new polymer electrolyte has shown some promising results. However a major revision is needed.

The reasons are the following:

1. to show the robustness of this new electrolyte, results at low temperature (e.g. -20 C) with different C-rate has to be given, but it is not found in the text or SI.

2. the proposed polymerization scheme is interesting, but still lack of some details. For example, what is the role played by FEC? It is mentioned that 10 wt% FEC is involved, but never mentioned in the proposed mechanism Fig. S2. This is quite confusing. More explanation is needed.

3. From Fig. S7, it looks like the polymer is liquid, maybe sticky. What is the measured viscosity?

4. The high ionic conductivity of this new polymer electrolyte is impressive. What if the FDMA and FMC relative ratio (5:3) is changed in the mixture?

Will it significantly affect the ionic conductivity, including the proposed polymerization process? Further discussion is needed. This help to understand this new class of electrolyte.

5. Fig. 2b shows a high Li transference number ~ 0.8 . Any explanation for this? How does it compare with the literature (e.g. Table S1)?

**6. pg. 5, line#94. "Formation of fluorine-rich interphases enhance the interfacial kinetics and stability."
==>Any citation supported this general statement?**

Other minor comments:

--several typo mistakes:

e.g. pg. 5, line#103. LiCO_3 ==> Li_2CO_3 ; specie (pg. 12, line#252)

**--Which C-rate is used for discharge-charge profiles (Fig. 2, 3).
Should be specified in the figure caption.**

--Fig. S1 is related to HOMO-LUMO orbitals, not energies.

Response Letter for Manuscript (Nature manuscript: NCOMMS-22-24260)

Tailoring quasi-solid polymer electrolyte for high-voltage solid-state Li metal batteries working at ultra-low temperatures

We sincerely appreciate the reviewer's endorsement of our work, and their insightful and constructive input. The manuscript has now been carefully revised, and the revisions are highlighted by **yellow color**. Our point-to-point responses to the comments are given in the following.

Reviewer #1 (Remarks to the Author):

This work reports a novel polymer electrolyte with high ionic conductivity and ionic transference number at low temperature. The Li||NCM811 battery assembled with such the polymer electrolyte presented high capabilities and stable long-term cycling at both room and low temperatures ($-20\text{ }^{\circ}\text{C}$) due to good electrochemical stability and fast kinetics. Moreover, the operation temperature of the batteries can be decreased to below $-30\text{ }^{\circ}\text{C}$, and even at $-58.3\text{ }^{\circ}\text{C}$. These results are very important, which can effectively promote the safety batteries to work at ultra-low temperatures. Therefore, this manuscript should be published in Nature Communications after addressing the following issues:

1. The authors briefly mentioned that FDMA with low melting point and a modest viscosity can greatly enhance the ionic transport in the electrolyte. The ionic conduction mode in the polymer electrolyte should be clarified.

Reply: Thanks for this comment. In a polymer electrolyte, ions are dissociated from counter-ions and coordinate with the electron-donor groups in the polymer host. Under the effect of an electric field, these cations tend to hop from one coordinating site to another. Such ion hopping is facilitated by either a segmental motion of the polymer chains or an ion-cluster-assisting function, in which temporary re-association with the counter-ions occurs before being re-solvated by electron donor groups in the polymer. It is thus generally believed that the ionic conduction in polymer electrolytes mainly occurs in the amorphous part of the polymer, while the crystalline part delivers very limited ion motion. When the temperature decreases below the glass transition temperature T_g , almost all polymer segments crystallize, and thus strong polymer-Li⁺

interactions lead to a large activation energy for the ion motion and a low ionic transference number (*Adv. Energy Mater.* 2018, 8, 1800703). Therefore, polymer electrolytes present low ionic conductivities with an order of 10^{-8} S cm⁻¹ at low temperatures.

Incorporating plasticizers into polymer matrices to form quasi-solid polymer electrolytes can greatly increase the volume of amorphous regions and promote segmental motion of polymer chains, which in turn realizes the high ionic conductivity and low-temperature operation. In quasi-solid polymer electrolytes, the transport of Li ions mainly occurs in the liquid regions containing dissolved Li salts (*Chem*, 2019, 5, 2326–2352). As shown in Fig. 1a, the commonly used liquid plasticizers show a high ice-point and viscosity. Once the temperature drops to -20 °C, the viscosity of liquids would increase and liquids are even frozen, which leads to a sharp decrease in the ionic conductivity. To ensure an acceptable ionic conductivity (at least 0.1 mS cm⁻¹) at low temperatures, novel electrolytes should contain a certain amount of liquid phase (amorphous regions) at a low temperature, e.g., -40 °C (*Adv. Mater.* 2022, 34, 2107899). Even at ultra-low temperatures, FDMA with a low melting point can maintain a liquid and an acceptable viscosity, thus significantly enhancing the low-temperature ionic conductivity.

2. At both low and room temperatures, the polymer electrolyte shows significantly enhanced Li⁺ transference number, which means that the transportation of anions is selectively restricted. The authors had better provide a more detailed explanation.

Reply: Thanks for this comment. The increased Li⁺ transference number can be attributed to the “assisted Li-ion diffusion” mechanism, in which a lithium ion is transported from one anion to another through binding sites, similar to the Li⁺ transport

behavior in an inorganic solid electrolyte (*J. Phys. Chem. B* 2018, 122, 2600–2609; *Angew. Chem. Int. Ed.* 2020, 59, 22194–22201). A Li-ion is firstly separated from a counter-ion and coordinated with a solvent/polymer molecule, and then hops and interacts with a different anion (Fig. R1) (*J. Phys. Chem. Lett.* 2016, 7, 4795–4801). Therefore, Li ions not only diffuse with solvent/polymer molecules but also hop between different Li-salt anions. The Li-ion hopping between anions also contributes to the high Li-ion diffusion in the electrolyte. The new electrolyte, containing strong electron withdrawing groups, such as C-F in FDMA, involves more anions in the Li⁺ solvation structure (*Nano Energy*, 2020, 73, 104786), which increases binding sites for the Li⁺ hopping. In addition, the strong interaction between anions of Li salt and ether oxygen groups in the electrolyte can limit the movement of anions. (*Adv. Sci.* 2017, 4, 1600377). Consequently, the diffusion coefficient of Li ions is larger than that of anions.

Fig. R1. Schematic illustrations of Li-ion hopping from anion 1 to anion 2 (*J. Phys. Chem. Lett.* 2016, 7, 4795–4801).

3. From the SEM images, it can be found that both the thicknesses of the Li deposition layer and the chunk size of Li on the surface clearly became smaller from room temperature to $-20\text{ }^{\circ}\text{C}$. Can the authors explain this phenomenon?

1). Small chunk size

Fig. R2. (a) Schematic diagram for the dependence of critical nucleus radius and areal nucleus density on the Li-deposition overpotential (*Nano Lett.* 2017, 17, 1132–1139). (b) Schematic illustration of Li nucleus size and nucleus density deposited on Cu at varying temperatures (*Nano Lett.* 2019, 19, 8664–8672).

Reply: In general, the Li nucleation process involves two critical parameters: the relaxation time (S) and the nucleation overpotential ($\Delta\eta$). The nucleus size is proportional to $(\Delta\eta)^{-1}$, while the density of nuclei is proportional to $(\Delta\eta)^3$ (as shown in the Fig. R2a) (*Nano Lett.* 2017, 17, 1132–1139). The critical nucleus radius is related to the deposition overpotential by the following equation:

$$r = 2\gamma V_m / (F |\Delta\eta|)$$

where r is the lithium nucleus radius, γ is the surface energy of the Li/electrolyte

interface, V_m is the molar volume of Li, F is Faraday's constant and $\Delta\eta$ is the nucleation overpotential. According to this theory concerning the initial stages of nucleation, a decreased temperature leads to a larger overpotential, which contributes to a decreased nucleus radius and a higher nucleation density (Fig. R2b) (*Nano Lett.* 2019, 19, 8664–8672; *Joule* 2020, 4, 1–16). Moreover, the slow mobility of Li ions caused by a low temperature leads to slow Li migration to separated nucleus sites that grow into small Li nuclei. Therefore, smaller Li particles are nucleated at low temperatures after further plating.

1). Smaller thickness

Fig. R3. Schematic illustrating the mechanism of Li nuclei generation and growth (*Joule* 2020, 4, 1–16).

Reply: Thanks for this comment. A larger nucleus radius and smaller nucleation density result in the formation of a thicker Li deposition layer at elevated temperatures (*J. Mater. Chem. A* 2018, 6, 4629–4635). As shown in Fig. R3, the high mobility of Li ions caused by a higher temperature facilitates rapid Li migration to separated nucleus sites that grow into large and sparse Li nuclei that fuse with their neighbors late. Finally, a dense and thicker Li deposition layer is generated after further plating. What's more,

temperature can affect the rate of the decomposition reaction of Li salts and solvents, and thus high reaction rates at higher temperatures result in a thicker SEI (*ACS Energy Lett.* 2020, 5, 2411–2420).

4. It is not clear whether the measured ionic conductivity is due to the polymer electrolyte with a PE separator or only the polymer electrolyte. The authors should give details of the measurement method.

Fig. R4. Schematic illustration of the EIS measurement setup.

Reply: The measured ionic conductivities are due to only the polymer electrolyte. The ionic conductivity and activation energy were obtained from EIS using a transparent glass bottle containing the polymer electrolyte with two pieces of stainless steel inserted inside electrolyte (as shown Fig. R4); the tested temperatures ranged from -20 to 60 °C, and the frequency range was from 5 MHz to 1 Hz. The ionic conductivities (σ) of the polymer electrolyte were calculated according to the equation:

$$\sigma = d/SR$$

where d (cm) is thickness of the polymer electrolyte, S (cm²) is the effective contact area with the stainless steel, and R (Ω) is the resistance measured by EIS.

5. The authors need to explain how to calculate the N/P ratio.

Reply: Thanks for the helpful comment. According to the work of Li et al. (*Nat. Energy* 2021, 6, 495–505), $N/P = Q_A/Q_C$, where Q_A is the area capacity of the Li metal anode, and Q_C is the area capacity of the cathode. The area capacity of the Li metal anode Q_A can be calculated as $Q_A = Q_{Li}m_{Li}$, where Q_{Li} is the specific capacity of the Li metal, which is 3860 mAh g⁻¹, m_{Li} is the weight of the Li metal. For the 50 μ m-thick Li anode, m_{Li} is 2.5×10^{-3} g cm⁻², and the area capacity can be calculated to be ~ 9.65 mAh cm⁻²; then the N/P ratio of our battery is ~ 3.8 . The calculation equation has been provided in the Method section.

6. This manuscript only gives the LSV curve and CV curves of the polymer electrolyte.

I recommend the authors provide these data of the reference electrolyte as well.

Fig. S11. Positive linear sweep voltammogram (LSV) of Li||Pt cell to gauge the oxidation stability of both electrolytes.

Reply: As shown in Fig. S11, the reference electrolyte shows a low oxidation potential,

as evidenced by a rapid increase in current above ~ 4.2 V; in contrast, oxidative stability of the polymer electrolyte improves dramatically, and no noticeable oxidative current is observed until 5.6 V.

According to the cyclic voltammetry (CV) tests (Fig. S12), it is observed that the reference electrolyte exhibits two additional reduction peaks at ~ 1.4 V and ~ 0.5 V, which represent the reduction of LiPF_6 salts and carbonate solvents, respectively (*Angew. Chem. Int. Ed.* 2020, 59, 2219). In contrast, the polymer electrolyte demonstrates much slighter reduction peaks, indicating no obvious side-reaction between the polymer electrolyte and LMA.

The relevant results and discussions have been provided in the manuscript and the Supporting information.

Fig. S12. Cyclic voltammetry (CV) curves of both electrolytes in Li||Pt cells.

7. Some electrochemical tests were performed under different conditions with respect to cathode loading, temperature, C-rate, etc., which might be confusing for readers. Some data, such as cathode loading, temperature, and C-rate, on each figure, should be presented.

Reply: Thanks for the comments, the relevant information, including cathode loading, temperature, and C-rate, have been specified on the figures or in the figure captions.

8. The authors suggested that amorphous $\text{Li}_x\text{BO}_y\text{F}_z$ can mechanically accommodate the large volume change of the electrode; however, detailed explanation is lack. Can the authors elaborate?

Reply: During the initial charge-discharge process, LiDFOB is rapidly reduced/oxidized on the electrode surface to construct $\text{Li}_x\text{BO}_y\text{F}_z$ -rich CEI/SEI. These processes exclude high temperature calcination so that the products are inclined to deliver amorphous state (*J. Am. Chem. Soc.* 2021, 143, 16768–16776). The amorphous property leads to a high-plasticity SEI/CEI that is coated on the surface of electrodes, which can buffer lattice distortion, extreme volume change during the prolonged electrochemical process (*Adv. Energy Mater.* 2019, 9, 1900626; *Angew. Chem. Int. Ed.* 2018, 57, 1505–1509; *Angew. Chem. Int. Ed.* 2020, 59, 6585–6589).

Reviewer #2 (Remarks to the Author):

The authors present a polymer-based electrolyte which strategically influences the composition of the SEI to promote low temperature operation of a lithium ion battery utilizing a lithium metal anode. This manuscript is of interest to the scientific community and suggested for publication after a number of major revisions are addressed.

1) Please define all acronyms upon first introduction in the manuscript

Reply: Thanks for the valuable comment. All acronyms are now defined upon first introduction in the manuscript.

2) The polymer electrolyte described herein is most likely a gel polymer electrolyte. Please describe the electrolyte as a gel polymer electrolyte within the manuscript or provide evidence that there are no liquid electrolyte domains within a solid polymer matrix in the electrolyte described in this manuscript.

Reply: Thanks for the suggestion. We correct the polymer electrolyte as “quasi-solid polymer electrolyte”. The interpretations are in the following:

In the case of this new electrolyte, before polymerization, TXE-FDMA-FEC-LiDFOB precursor is a flowable and transparent liquid, and the molecular weight of the precursor is less than 150 (TXE: 90.08, FDMA: 141.09, FEC: 106.05, and LiDFOB: 143.77). After polymerization, the original flowable liquid solution precursor turns into a white solid-like product and losses the fluidity (Fig. S7). The molecular number of our polymer is ~137778 (Fig. S6 and Table S1). Obviously, the electrolyte shows solid characteristics.

However, there are 29.4 wt.% free solvents within the electrolyte, so that it should

be a quasi-solid (Before drying, the weight of the pristine electrolyte is 2.11 g; after fully dried at 100 °C in vacuum, the residual is 1.49 g in weight; therefore, the content of free solvents is less than 29.4%).

Generally, polymer electrolytes for Li-based batteries can be divided into three major categories: solvent-free polymer electrolytes, quasi-solid/gel polymer electrolytes, and composite polymer electrolytes (*Chem 5, 2019, 2326–2352*). As an important member of the polymer electrolyte family, quasi-solid polymer electrolyte can well describe this new electrolyte.

Fig. S6. GPC curves of prepared electrolytes with different TXE-FDMA ratios.

Table S1. GPC results of prepared electrolytes using different TXE and FDMA ratios, where M_n is the number-average molecular weight, M_w is the weight-average molecular weight, M_P is the molecular weight of the highest peak, and PDI is the polydispersity.

Samples	M_n (Daltons)	M_w (Daltons)	M_P (Daltons)	PDI
5:1	16334	31011	31220	1.90
5:2	15804	31239	32692	1.98
5:3	13778	28987	30262	2.10
5:5	12295	29477	30863	2.39
3:5	11587	27898	28780	2.41

Fig. S7. Optical images of the precursor and the polymer electrolyte.

3) The following statement in the introduction requires correction or clarification:

"In contrast, LiDFOB exhibits the lowest LUMO, demonstrates a high electron affinity and can be expected to be reduced firstly, hence it can dominate the formation of the outer-layer SEI; while FDMA with strong electron-withdrawing $-CF_3$ and $-N-$ groups shows the second-lowest LUMO energy, it can contribute to form LiF-rich and Li_2CO_3 -less inner-layer SEI (Fig. 1d)."

If LiDFOB were reduced first, this would contribute to the inner-layer SEI and FDMA would contribute to the outer layer SEI.

Reply: Thanks for the valuable comment. The formation of the dual-layered film on the Li metal anode is illustrated in Fig. R5. LiDFOB exhibits the lowest LUMO energy and demonstrates a high electron affinity, leading to its prior reduction to form a $Li_xBO_yF_z$ -rich layer at the lower charge-voltage (Stage 1). The formation process of $Li_xBO_yF_z$ excludes high temperature calcination, so that the $Li_xBO_yF_z$ film is inclined to deliver amorphous state (*J. Am. Chem. Soc.* 2021, 143, 16768–16776). Due to the ionic

conduction and electrical insulation, electrons from the anode cannot transport across the $\text{Li}_x\text{BO}_y\text{F}_z$ layer, limiting the reduction of the FDMA solvent on the surface (*Nat Energy*, 2020, 5, 386–397). Subsequently, FDMA infiltrates and passes through the $\text{Li}_x\text{BO}_y\text{F}_z$ layer, along with Li^+ , and then it can be reduced to form the LiF-rich film under the $\text{Li}_x\text{BO}_y\text{F}_z$ layer by the acceptance of the electrons from the anode (Stage 2). Therefore, LiF-rich components tend to enrich in the inner part, while $\text{Li}_x\text{BO}_y\text{F}_z$ components are distributed in the outer part in the dual-layered interface structure. This dual-layered SEI shielding layer is in a dynamically equilibrated state to maintain a controllable thickness and stable surface electrochemistry throughout the charge-discharge process (Stage 3) (*Nat. Energy* 2018, 3, 739–746).

Fig. R5. Schematic diagram of the dual-layered film formation on the Li metal anode.

4) Remove unnecessary articles such as "the" in "the Raman spectroscopy" and "the nuclear magnetic resonance" and "the mass spectrometry".

Reply: Thanks for the suggestion. We have removed "the" from the relevant positions.

5) The operation of the pouch cell at -58.3 C needs to be conducted in a thermally equilibrated temperature bath. The use of dry ice in its solid state provides insufficient thermal contact to provide confidence in the overall pouch cell temperature. The use of a slush bath is advised to provide adequate thermal contact and confidence in the

equilibrated temperature of the pouch cell. See DOI: 10.1002/adv.202106032 for an example of this.

Fig. 3e. The Li||NCM811 pouch cell using the designed polymer electrolyte was powering an electric fan at about $-48.2\text{ }^{\circ}\text{C}$.

Reply: Thanks for this helpful suggestion. A new experiment was conducted to evaluate the practical application at low temperatures. To achieve an equilibrated temperature, the pouch cell was conducted in a dry-rice/ethanol slush bath. As shown in Fig. 3e and Movie S1, the pouch cell can power an electronic fan at about $-48.2\text{ }^{\circ}\text{C}$.

6) Please highlight the equipment used to ensure temperature stability during cycling studies in the experimental section.

Reply: All cells were tested in an environmental test chamber to maintain a stable temperature environment (MT3065, $-60\sim 150\text{ }^{\circ}\text{C}$, Guangzhou-GWS Environmental Equipment Co., Ltd.).

Reviewer #3 (Remarks to the Author):

This work is interesting and timely. The new polymer electrolyte has shown some promising results. However a major revision is needed.

The reasons are the following:

1. to show the robustness of this new electrolyte, results at low temperature (e.g. $-20\text{ }^{\circ}\text{C}$) with different C-rate has to be given, but it is not found in the text or SI.

Reply: The rate performance of Li||NCM811 with the designed electrolyte at $-20\text{ }^{\circ}\text{C}$ is shown in Fig. S18 (Supporting Information). The Li||NCM81 cell delivers highly reversible specific capacities of 147.3, 122.5, 111.9, 95.1 mAh g^{-1} at 0.1, 0.2, 0.3, and 0.5 C, respectively. When the rate resets to 0.1 C, the cell recovers to a considerable capacity of 150.8 mAh g^{-1} . These results indicate that the designed electrolyte has good robustness property and superior rate recoverable performance at low temperatures.

Fig. S18. **a.** Rate performance of the Li||NCM811 cell with the designed electrolyte at $-20\text{ }^{\circ}\text{C}$, and **b.** corresponding charge-discharge voltage profiles at different C-rates.

2. the proposed polymerization scheme is interesting, but still lack of some details. For example, what is the role played by FEC? It is mentioned that 10 wt% FEC is involved, but never mentioned in the proposed mechanism Fig. S2. This is quite confusing. More explanation is needed.

Reply: Firstly, FEC shows very low HOMO level, therefore, the addition of FEC significantly increases the oxidation potential of the electrolyte (*Nat. Commun.* 202011, 4188). Secondly, FEC can regulate the solvation behavior and the SEI formation process, thus stabilizing the electrodes (*Angew. Chem., Int. Ed.* 2018, 57, 5301; *ACS Energy Lett.* 2020, 5, 2411). Due to the competitive effect, the presence of organic solvents/residual monomers/low-molecule polymers in the first solvation shell diminishes, while the percentage of anions and FEC increases upon adding FEC, thus more anion-derived SEI ($\text{Li}_x\text{BO}_y\text{F}_z$ -rich) can be achieved. Thirdly, FEC can stabilize the electrolyte, because the residual monomers/low-molecule polymers in the electrolytes are unstable at a low potential (*Chem*, 2019, 5, 2326–2352, *Chem. Eur. J.* 2021, 27, 15842–15865).

3. From Fig. S7, it looks like the polymer is liquid, maybe sticky. What is the measured viscosity?

Fig. S7. Optical images of the precursor and the polymer electrolyte.

Reply: Thanks for your helpful comments. Before polymerization, TXE-FDMA-FEC-

LiDFOB precursor is a flowable and transparent liquid. After polymerization, the originally flowable liquid precursor turns into a white solid-like product, and loses the fluidity, and the molecular weight increase from less than 150 to 13788 (Table S1). More importantly, the electrolyte remains solid even at an elevated temperature of 100 °C.

Owing to its solid-like characteristics, we cannot measure the viscosity. The relevant results and discussion have been provided in Fig. S8 (Supporting information).

4. The high ionic conductivity of this new polymer electrolyte is impressive.

What if the FDMA and FMC relative ratio (5:3) is changed in the mixture?

Will it significantly affect the ionic conductivity, including the proposed polymerization process? Further discussion is needed. This help to understand this new class of electrolyte.

Fig. R8. Room-temperature ionic conductivities of the polymer electrolytes with various ratios of TXE and FDMA.

Table S1. GPC results of prepared electrolytes using different TXE and FDMA ratios, where M_n is the number-average molecular weight, M_w is the weight-average molecular weight, M_P is the molecular weight of the highest peak, and PDI is the polydispersity.

Samples	M_n (Daltons)	M_w (Daltons)	M_P (Daltons)	PDI
5:1	16334	31011	31220	1.90
5:2	15804	31239	32692	1.98
5:3	13778	28987	30262	2.10
5:5	12295	29477	30863	2.39
3:5	11587	27898	28780	2.41

To achieve satisfactory conductivity values, liquid plasticizers are inevitably required. When the mass ratios of TXE and FDMA are 5:1 and 5:2, they have relatively low ionic conductivities at room temperature. With the increase of the ratio of FDMA, the ionic conductivity of the electrolyte rapidly increases (Fig. S8).

As listed in Table S1, M_n of polymers generally decreases with the increment of the FDMA content. The higher content of FDMA retards the reaction rate, causing polymer chains to grow more slowly, which would lower polymer molecular weight. At the TXE-FDMA mass ratios of 5:5 and 3:5, the electrolytes contain high content of low-molecule polymers. High content solvents in the electrolytes not only cause poor thermal/mechanical stability and safety hazards such as fire and explosion during thermal runaway, but also greatly deteriorates the electrode-electrolyte interfaces (*Chem*, 2019, 5, 2326–2352).

In view of the balance between the thermal/mechanical/(electro)chemical stability and the ionic conductivity, the electrolyte with a TXE-FDMA ratio of 5:3 exhibits a suitable molecular weight ($M_n = 13778$), a relatively narrow polydispersity (2.1) and an

acceptable room-temperature ionic conductivity (2.5 mS cm^{-1}). Hence, we finally choose the polymer electrolyte with the TXE-FDMA ratio of 5:3 as the electrolyte for further investigations.

5. Fig. 2b shows a high Li transference number ~ 0.8 . Any explanation for this?

How does it compare with the literature (e.g. Table S1)?

Fig. R6. Schematic illustrations of Li-ion hopping from anion 1 to anion 2 (*J. Phys. Chem. Lett.* 2016, 7, 4795–4801).

Reply: Thanks for this comment. The increased Li^+ transference number can be attributed to the “assisted Li-ion diffusion” mechanism, in which a lithium ion is transported from one anion to another through binding sites, similar to the Li^+ transport behavior in an inorganic solid electrolyte (*J. Phys. Chem. B* 2018, 122, 2600–2609; *Angew. Chem. Int. Ed.* 2020, 59, 22194–22201). A Li-ion is firstly separated from a counter-ion and coordinated with a solvent/polymer molecule, and then hops and interacts with a different anion (Fig. R1) (*J. Phys. Chem. Lett.* 2016, 7, 4795–4801). Therefore, Li ions not only diffuse with solvent/polymer molecules but also hop between different Li-salt anions. The Li-ion hopping between anions also contributes to the high Li-ion diffusion in the electrolyte. The new electrolyte, containing strong

electron withdrawing groups, such as C-F in FDMA and FEC, involves more anions in the Li⁺ solvation structure (*Nano Energy*, 2020, 73, 104786), which increases binding sites for the Li⁺ hopping. In addition, the strong interaction between anions of Li salt and ether oxygen groups in the polymer can limit the movement of anions (*Adv. Sci.* 2017, 4, 1600377). Consequently, the diffusion coefficient of Li ions is larger than that of anions.

As discussed above, the Li⁺ transference number of our electrolyte is higher than those reported in literatures, as summarized in Table S2.

6. pg. 5, line#94. "Formation of fluorine-rich interphases enhance the interfacial kinetics and stability."

==>Any citation supported this general statement?

Reply: Archer and co-workers explored the activation energy barrier for the Li diffusion at the interphase between electrolyte and Li-metal anode by the joint density functional theoretical calculations, and demonstrated that the surface diffusivity of Li⁺ increases by more than two orders of magnitude with the presence of fluorides SEI (*Nat. Mater.* 2014, 13, 961–969). Zhang's group suggested that the fluorinated SEI renders uniform spatial diffusion of Li ions and guides the deposited Li into an ordered and aligned columnar structure, thus suppressing the dendrite growth. Additionally, highly homogeneous fluorinated SEIs largely minimize the side-reaction between the electrolyte and the Li metal anode during long-term cycling (*Joule*, 3, 2647–2661).

The relevant references have been cited (Reference 32, 33) to support the statement.

Other minor comments:

--several typo mistakes:

e.g. pg. 5, line#103. $\text{LiCO}_3 \implies \text{Li}_2\text{CO}_3$; specie (pg. 12, line#252)

Reply: Thanks for this helpful comment. We have corrected this typo mistake.

--Which C-rate is used for discharge-charge profiles (Fig. 2, 3).

Should be specified in the figure caption.

Reply: Thanks for this helpful comment. The C-rate is now specified on these figures or in figure captions.

--Fig. S1 is related to HOMO-LUMO orbitals, not energies.

Reply: Thanks for this helpful comment. We have corrected this mistake.

REVIEWER COMMENTS

Reviewer #1 (Remarks to the Author):

Accept

Reviewer #2 (Remarks to the Author):

The modifications to the manuscript are satisfactory with the exception of one aspect. In the response to reviewer's comments the authors state "The formation of the dual-layered film on the Li metal anode is illustrated in Fig. R5. LiDFOB exhibits the lowest LUMO energy and demonstrates a high electron affinity, leading to its prior reduction to form a LixBOyFz rich layer at the lower charge-voltage (Stage 1). The formation process of LixBOyFz excludes high temperature calcination, so that the LixBOyFz film is inclined to deliver amorphous state (J. Am. Chem. Soc. 2021, 143, 16768–16776). Due to the ionic conduction and electrical insulation, electrons from the anode cannot transport across the LixBOyFz layer, limiting the reduction of the FDMA solvent on the surface (Nat Energy, 2020, 5, 386–397). Subsequently, FDMA infiltrates and passes through the LixBOyFz layer, along with Li+, and then it can be reduced to form the LiF-rich film under the LixBOyFz layer by the acceptance of the electrons from the anode (Stage 2). Therefore, LiF-rich components tend to enrich in the inner part, while LixBOyFz components are distributed in the outer part in the dual-layered interface structure."

The references provided and experimental results in provided in the manuscript do not provide adequate experimental results to verify confirm this mechanism. Please modify the manuscript to indicate to the readers that this is a proposed mechanism. (i.e. "we believe this the be mechanism, but further experimental validation is required.")

Reviewer #3 (Remarks to the Author):

The revised manuscript quality has improved significantly. I think it can be published and further revision is required.

Response Letter for Manuscript (Nature manuscript: NCOMMS-22-24260A)

Tailoring quasi-solid polymer electrolyte for high-voltage solid-state Li metal batteries working at ultra-low temperatures

We sincerely appreciate the reviewer's endorsement of our work, and their insightful and constructive input. The manuscript has now been carefully revised, and the revisions are highlighted by **yellow color**. Our point-to-point responses to the comments are given in the following.

Reviewer #1 (Remarks to the Author):

Accept

Reply: We sincerely appreciate the reviewer's endorsement of our work.

Reviewer #2 (Remarks to the Author):

The modifications to the manuscript are satisfactory with the exception of one aspect. In the response to reviewer's comments the authors state

"The formation of the dual-layered film on the Li metal anode is illustrated in Fig. R5. LiDFOB exhibits the lowest LUMO energy and demonstrates a high electron affinity, leading to its prior reduction to form a LixBOyFzrich layer at the lower charge-voltage (Stage 1). The formation process of LixBOyFz excludes high temperature calcination, so that the LixBOyFz film is inclined to deliver amorphous state (J. Am. Chem. Soc. 2021, 143, 16768–16776). Due to the ionic conduction and electrical insulation, electrons from the anode cannot transport across the LixBOyFz layer, limiting the reduction of the FDMA solvent on the surface (Nat Energy, 2020, 5, 386–397). Subsequently, FDMA infiltrates and passes through the LixBOyFz layer, along with Li+, and then it can be reduced to form the LiF-rich film under the LixBOyFz layer by the acceptance of the electrons from the anode (Stage 2). Therefore, LiF-rich components tend to enrich in the inner part, while LixBOyFz components are distributed in the outer part in the dual-layered interface structure."

The references provided and experimental results in provided in the manuscript do not provide adequate experimental results to verify confirm this mechanism. Please modify the manuscript to indicate to the readers that this is a proposed mechanism. (i.e. "we believe this be the mechanism, but further experimental validation is required.")

Reply: Thanks for this helpful comment. We totally agree with the reviewer; a sentence "we believe this be the mechanism, but further experimental validation is required" has been added in the revised manuscript.

Reviewer #3 (Remarks to the Author):

The revised manuscript quality has improved significantly. I think it can be published and further revision is required.

Reply: We sincerely appreciate the reviewer's endorsement of our work.